



# Assessing the influence of ocean alkalinity enhancement on a coastal phytoplankton community

Aaron Ferderer[1], Zanna Chase[1], Fraser Kennedy[1], Kai G. Schulz[2], Lennart T. Bach[1]

[1]Institute for Marine and Antarctic Studies, Ecology & Biodiversity, University of Tasmania, Hobart, TAS, Australia.

[2]Centre for Coastal Biogeochemistry, School of Environment, Science and Engineering, Southern Cross University, Lismore, NSW, Australia.

*Correspondence to:* Aaron Ferderer (aaron.ferderer@utas.edu.au)



**Abstract.** Ocean alkalinity enhancement (OAE) is a proposed method to counteract climate change by increasing the alkalinity of the surface ocean and thus the chemical storage capacity of seawater for atmospheric $CO_2$. The impact of OAE on marine ecosystems, including phytoplankton communities which make up the base of the marine food web, are largely unknown. To investigate the influence of OAE on phytoplankton communities we enclosed a natural plankton community from coastal Tasmania for 22 days in nine microcosms

during a spring bloom. Microcosms were split into three groups, (1) the unperturbed control, (2) the unequilibrated treatment where alkalinity was increased (+495 ±5.2 µmol/kg) but seawater $CO_2$ was not in equilibrium with atmospheric $CO_2$, and (3) the equilibrated treatment where alkalinity was increased (+500 ±3.2 µmol/kg) and seawater $CO_2$ was in equilibrium with atmospheric $CO_2$. Both treatments have the capacity to increase the inorganic carbon sink of seawater by 21%. We found that simulated OAE had significant but

generally moderate effects on various groups in the phytoplankton community and on heterotrophic bacteria. More pronounced effects were observed for the diatom community where silicic acid draw-down and biogenic silica build-up were reduced at increased alkalinity. Observed changes in phytoplankton communities affected the temporal trends of key biogeochemical parameters such as the organic matter carbon-to-nitrogen ratio. Interestingly, the unequilibrated treatment did not have a noticeably larger impact on the phytoplankton (and

heterotrophic bacteria) community than the equilibrated treatment, even though the changes in carbonate chemistry conditions were much more severe. This was particularly evident from the occurrence and peak of the phytoplankton spring bloom during the experiment, which was not noticeably different from the control. Altogether, the inadvertent effects of increased alkalinity on the coastal phytoplankton communities appear to be justifiable, relative to the enormous climatic benefit of increasing the inorganic carbon sink of seawater by 21%.

**1 Introduction**

Keeping global warming below 2°C requires drastic and rapid emission reductions. In addition, a portfolio of Carbon Dioxide Removal (CDR) methods is required to extract several hundred gigatonnes of $CO_2$ from the atmosphere and store it safely in other carbon reservoirs for thousands of years (Rogelj et al., 2018). However, few CDR methods have been proven to work at this scale and all have potential side effects for the Earth system

(Fuss et al., 2018).

One potential method of CDR from the marine portfolio is Ocean Alkalinity Enhancement (OAE). The idea behind OAE is to increase the chemical storage capacity of the ocean for atmospheric $CO_2$ by adding proton-neutralizing substances to the surface ocean (Kheshgi, 1995). This is measurable as an enhancement of seawater

alkalinity, the name-giving process behind OAE. Enhanced alkalinity causes a shift in the inorganic carbon speciation in seawater, from carbon dioxide ($CO_2$) to bicarbonate ($HCO_3^-$) and carbonate ($CO_3^{2-}$) thereby "making new space" for additional atmospheric $CO_2$ to be absorbed (Hartmann et al., 2013). In addition to generating CDR, the absorption of protons through OAE counteracts ocean acidification (OA), which is considered an environmental threat for a range of marine ecosystems (Doney et al., 2020).


OAE can be achieved through a variety of approaches (Renforth and Henderson, 2017). Most of these approaches are either directly or indirectly linked to the chemical weathering of minerals, which neutralize protons when they dissolve. The simplest approach is to extract suitable minerals via mining, grind those



minerals into a powder, and distribute them over land or ocean surfaces where they can dissolve in aqueous

media over days to decades (Feng et al., 2017; Taylor et al., 2016). When applied on humid land surfaces, this CDR method is usually referred to as "Enhanced Weathering" (Schuiling and Krijgsman, 2006). Here, alkalinity and other mineral dissolution products associated with the ground minerals such as dissolved silicate or trace metals would primarily affect terrestrial ecosystems but ultimately wash into the oceans via rivers (Köhler et al., 2010). When ground minerals are added directly to the surface ocean (OAE), dissolution products, such as trace

metals, affect ocean biota immediately (Bach et al., 2019). In both cases, the release of alkalinity and other dissolution products is highly dependent on the applied source mineral (Renforth and Henderson, 2017). Mineral weathering can be further accelerated when ground minerals are dissolved in electrolysis cells for hydrogen production (Rau et al., 2013). Here, hydrogen serves as a valuable co-product to CDR, with alkalinity and other dissolution products still being formed, and requiring deposition in the environment where they

potentially affect biota. Another approach is the electrodialytic separation of water into acid and alkalinity (de Lannoy et al., 2018). Here, alkalinity (in the form of hydroxide) is maintained in the surface ocean enabling CDR (de Lannoy et al., 2018). The acid can be utilized commercially (e.g. as hydrochloric acid), stored in geological reservoirs underground, or pumped into the deep ocean where it is partially neutralized through the dissolution of carbonate sediments (Tyka et al., 2022). The advantage of this approach is that it does not directly

depend on mineral weathering so that mineral supply chains become redundant and no dissolution co-products (e.g. trace metals) are released into the environment (Tyka et al., 2022).

It is currently not possible to predict which of the approaches described above will be implemented in the future. Furthermore, it is unclear how ocean ecosystems would be affected by OAE, as each method differs in the

quality and quantity of released dissolution products. However, what all approaches have in common is the intentional change in carbonate chemistry via the addition of alkalinity. It is therefore an important first step to assess if increased seawater alkalinity constitutes a threat to the environment or not (Bach et al., 2019).

This study investigates, for the first time, if and how the changes in carbonate chemistry due to OAE influences

coastal phytoplankton communities. More explicitly, we compared the effects of two different alkalinity addition scenarios. Scenario one assumes that the surface ocean is in equilibrium with the overlying atmosphere so that the fugacity of $CO_2$ ($fCO_2$) in seawater is equal to that in the overlying atmosphere (the "equilibrated treatment"). Scenario two assumes that alkalinity is added but atmospheric $CO_2$ has not yet been absorbed by the perturbed seawater (the "unequilibrated treatment"). This second scenario is highly relevant because $CO_2$

equilibration can take months to years (Jones et al., 2014) and carbonate chemistry changes are substantially more pronounced in this unequilibrated transient state that occurs after the alkalinity addition (Bach et al., 2019).

The treatments were tested with a natural plankton community from coastal Tasmania and compared to an

unperturbed control. The communities were enclosed in nine identical microcosms in late winter with high nutrient concentrations naturally available. Our goal was to study OAE effects during the spring bloom, an ecologically and biogeochemically important event in the seasonal cycle with the highest biomass accumulation rates during the year.



## 2 Methods

### 2.1 Microcosm setup and mixing methods


This experiment made use of Kegland ® Fermzilla, conical uni tank fermenters as microcosms for the monitoring of coastal phytoplankton communities (Fig. 1c). Each microcosm consisted of a ~55 litre PET conical tank and a butterfly dump valve connected to a 1 litre collection container (sediment collection cup) (Fig. 1c). Microcosms were heated from the base of the conical tank using two 30-watt heat-belts to induce

convective mixing. This prevented the plankton community from sinking out of the water column in a non-invasive way (i.e. without a stirrer; Fig. 1c). To test the efficiency of the convective mixing, we filled eight microcosms with ~50 L of seawater sourced from the Derwent Estuary and placed them in a temperature-controlled room set to 8°C. This temperature was selected so that once heating was applied, the water temperature in microcosms would be within the range observed in the Derwent estuary during late winter (12-

14°C). Once the enclosed seawater had reached thermal equilibrium the heating on four of the microcosms was turned off. Thirty minutes later, 2.5ml of blue dye (food colouring) was added to all eight microcosms, four with no heating and four with heating applied (Fig. 1e). The blue dye was added with a pipette to the uppermost ~5 cm of seawater enclosed in microcosms. The rate of mixing within microcosms was then assessed by regularly measuring the absorbance of water samples taken from each microcosm in a spectrophotometer at 630 nm.

Samples were carefully taken from the top of each microcosm using a pipette at a depth of ~5cm below the water surface. After three hours all microcosms were manually mixed with a plastic stirrer to ensure homogeneity. After mixing, the absorbance was measured an additional three times and used as a reference for a homogeneously mixed solution.

Microcosms that had the convection system switched on were well mixed after approximately 30 minutes (Fig. 1d). In contrast, the "no-convection" microcosms where the convection system was switched off remained relatively un-mixed, expressed as variable dye concentrations measured with the spectrophotometer (Fig. 1d). The variability in absorbance was consistent with our observations, as filaments of high dye concentration were observed inside the no-convection microcosms until they were manually mixed (Supplementary video 1). It is

important to note that there was residual convective mixing within the no-convection microcosms, as the convection system was switched off only 30 minutes before the experiment allowing residual heat to enter the system (supplementary video 1). The rapid mixing induced via convection as observed in the dye experiment was confirmed by observations during the experiment, with large aggregates suspended in the water column failing to sink into the sediment trap (Fig. 1f, supplementary video 2). Thus, the convection mechanism used

here is an effective and non-invasive method to keep plankton in suspension and prevent the unrealistic sinking of particles.

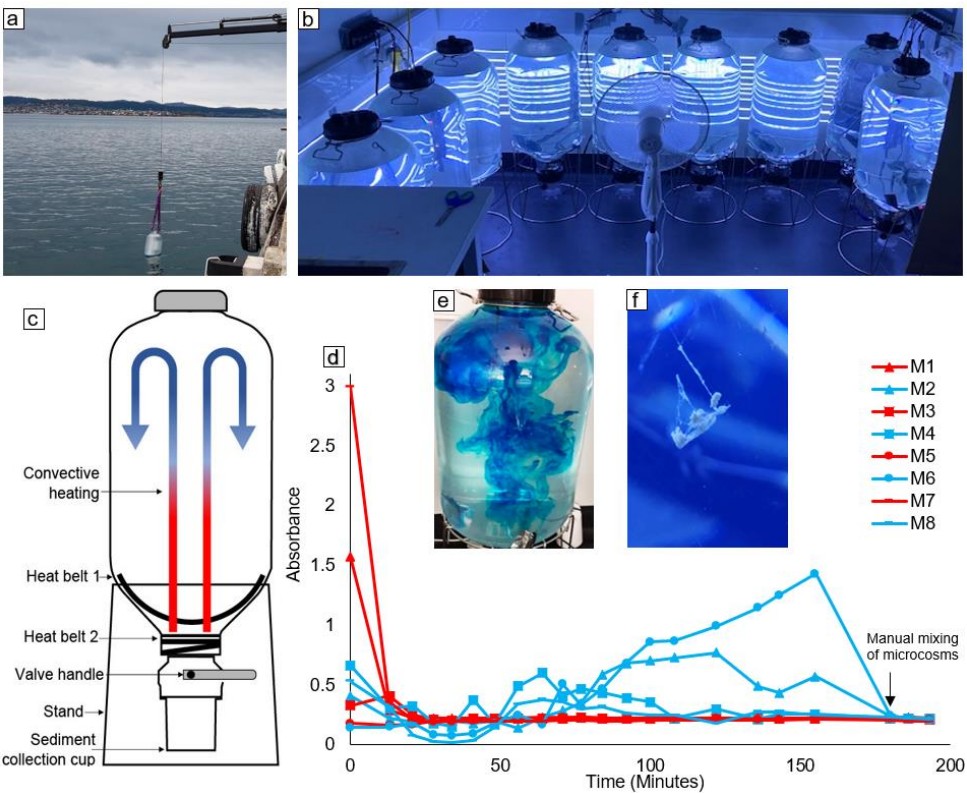

**Figure 1.** a) Method and location of microcosm filling, b) experimental setup, c) schematic diagram of the microcosms used in this study, d) results of the convective mixing test (microcosms with convective mixing are indicated by red lines and no-convection microcosms indicated by blue lines), e) microcosm with dye addition for assessment of convective mixing, and f) formation of an aggregate within a microcosm.

**2.2 Enclosure of phytoplankton communities, treatment manipulation, and initiation of the experiment**

Nine microcosms were filled with seawater from the Derwent Estuary (August 2021) outside the University of Tasmania Institute for Marine and Antarctic Studies building (42.53095S, 147.20101E, Fig. 1a). We refrained from pumping the water into the microcosms as this may harm organisms and alter the plankton community composition. Instead, microcosms were gently filled from the base up (similar to a Niskin sampler) by lowering microcosms one at a time into the water, approximately five meters out from the edge of the wharf (Fig. 1a). Water was filtered through a 2 mm mesh screen attached to the top and base of microcosms prior to filling. The base of each microcosm was submerged to a depth of ~1 meter below the surface and the base closed using a rope attached to the valve handle. Sediment collection cups were then attached to all microcosms with the valve closed. The filling procedure lasted less than 30 minutes, ensuring enclosure of similar water masses. All



microcosms were weighed separately before and after the filling procedure and contained volumes ranging between 55.2 to 55.9 L.


Microcosms were then transported to a temperature-controlled room set to 8°C (±2°C) and heat belts attached as per the methods outlined in Sect. 2.1 (Figs. 1b, c). To simulate natural light conditions, ten LED light strips were installed in the room providing a cool white light source with approximately 200 µmol photons $m^{-2}$ $s^{-1}$ inside each microcosm on a 12:12 light:dark cycle. Light intensity was measured in the centre of each microcosm with a quantum light meter (LI-COR Biosciences, Lincoln, USA). Due to slight variations in temperature and


irradiance throughout the room, microcosms were rotated around the room once a day at ~1100 hrs (Fig. 1b). The temperature of the room was lowered from 8°C to 6.5°C over the course of the experiment to ensure temperature stability within the microcosms at 12-14°C (Fig. 2). This was necessary because the reduced volume of water within microcosms due to sampling caused an increase in heat input per volume via the heat belts so that the cooling from outside had to be increased. Salinity of the seawater enclosed was 34.5 as


measured with a 914 Metrohm salinometer.

Microcosms were split into three groups; a control (M1, M4, M7), which received no alkalinity manipulation, the "unequilibrated" group (M2, M5, M8) enriched with 500 µL of NaOH (Merck, Titripur) per litre, and the "equilibrated" group (M3, M6, M9) enriched with 423 µL of 1 molar $NaHCO_3$ solution (prepared by dissolving


8.401 g of $NaHCO_3$ (Sigma-Aldrich) in 100 mL of double-deionised water) per litre and 77 µL of NaOH (Merck, Titripur) per litre. The mixing ratio of $NaHCO_3$ and NaOH in the equilibrated group was determined with the carbonate chemistry calculation software seacarb (Gattuso et al., 2021) prior to the manipulation, assuming that the collected seawater had a total alkalinity of 2280 µmol/kg and the $fCO_2$ was in equilibrium with the atmosphere (~410 µatm). A more detailed description of the calculation of carbonate chemistry


conditions is provided in Sect. 2.4. The whole procedure lasted four hours and we consider the end of the manipulation as the beginning of the experiment.

**2.3 Seawater sampling and particulate matter analyses**

Samples were extracted from all microcosms between 0700-0900 hrs, however, sampling intervals varied


depending upon the parameter as indicated in Fig. 2. Prior to sampling, each microcosm was gently mixed in a circular motion five times, using a 60 cm plastic stirrer to ensure no sedimentation bias was introduced in the sampling (this was carried out as a precaution, even though preliminary tests with flow cytometry illustrated that homogenization was achieved with convective mixing alone, data not shown). Seawater was sampled from the microcosms using either a silicon tube (particulate matter) or a Tygon tube (nutrients, total alkalinity, flow


cytometry) and pumped directly into clean bottles (pre-rinsed with sample). Sampled volumes ranged between 125 ml-1250 ml, depending on the parameters assessed. Samples for dissolved inorganic nutrients (nitrate + nitrite, phosphate, and silicate) and total alkalinity were filtered through a syringe filter (0.2 µm, Millipore) to minimize biological processes. Nutrient concentrations were analysed within 5 hours after sampling (Sect. 2.4). Total alkalinity samples were stored at 6°C in the dark for 0-14 days until analysis (analyses described in Sect.


2.4).



Samples for chlorophyll *a*, biogenic silica (BSi), total particulate carbon (TPC), and nitrogen (TPN) were taken by filtration of 150-240 mL at a mild vacuum pressure of -200 mbar relative to the atmosphere. Blank filters (placed onto the filtration rack without filtering particles onto them) were prepared for all four parameters during each sampling day. TPC and TPN were filtered on pre-combusted (6 hours at 450°C) quartz fibre (QMA, Whatman) filters (nominal pore-size = 2.2 µm) and stored at -4°C in pre-combusted (6 hours at 450°C) glass petri dishes for 3-25 days. Prior to analysis, filters were dried at 60°C for 2 hours, packaged into tin foil, and analysed using a Thermo Finnigan EA 1112 Series Flash Elemental Analyser. Combustion of the pressed tin cups was achieved in high purity oxygen at 1000°C using tungstic oxide on alumina as an oxidising agent followed by copper wires as a reducing agent. The results were calibrated using a certified sulphanilamide standard. Please note that we conducted flow-cytometric test measurements where we filtered samples from the microcosms through the QMA filters to test if pico-phytoplankton (0.2-2 µm) would be retained on the filters. These measurements revealed that pico-phytoplankton did not pass through the QMA filters, thus the entire phytoplankton community was sampled (Fig. A1).

BSi was filtered on 3 µm nitrocellulose membrane filters which were then stored in plastic petri dishes for 51-73 days at -4°C until samples were analysed. For the analysis, BSi first needed to be converted into silicic acid. For this, filters were put into 60mL polypropylene vials filled with 0.1 molar NaOH solution, the vials were firmly closed, and heated for 135 minutes at 80°C in a temperature-controlled bath. Afterward, the vials were allowed to cool down to room temperature and the silicic acid concentration was measured photometrically following Hansen and Koroleff (1999).

Chlorophyll *a* samples were filtered through glass fibre filters (GF/F, nominal pore size = 0.7 µm). After filtration, filters were carefully folded, placed in 15 mL polypropylene tubes wrapped in aluminium foil, and immediately frozen and stored at -80°C. After extraction with 10 ml of methanol (100%) for 14 hours, samples were analysed fluorometrically on a Turner fluorometer following the acidification method outlined by Evans et al. (1987).

Samples for scanning electron microscopy (SEM) were taken by filtration of 30 mL at a mild vacuum pressure of -200 mbar relative to the atmosphere through 0.2 µm polycarbonate filters and dried for 2 hours at 60°C in a desiccator. Prior to analysis, samples were glued onto aluminium stubs and sputtered with gold-palladium. Samples were analysed in a Hitachi SU-70 analytical field emission scanning electron microscope.





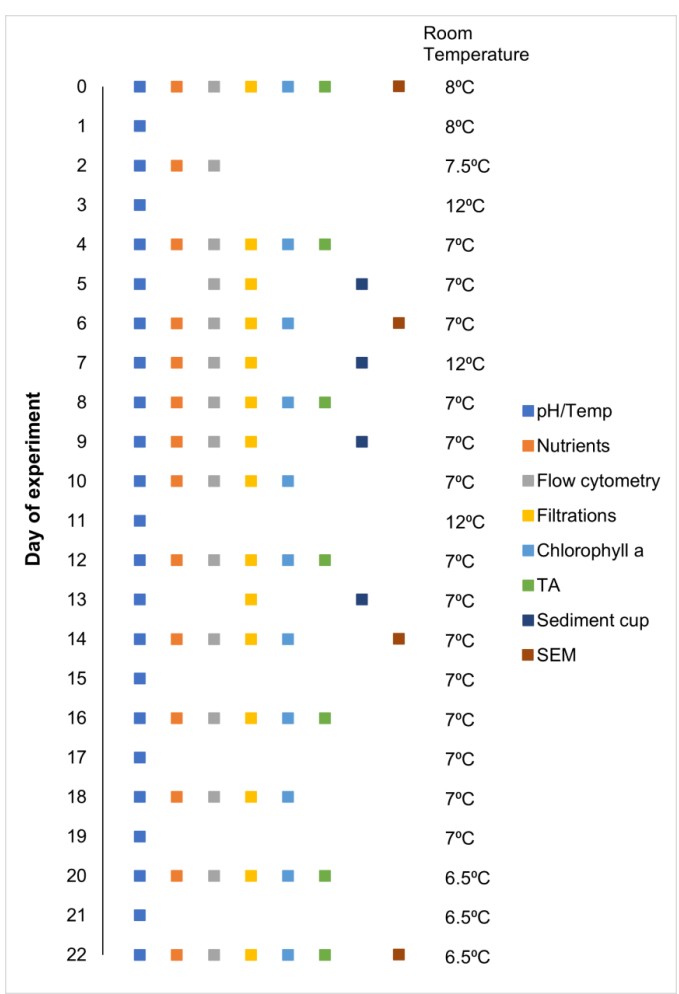

**Figure 2.** Sampling schedule for given parameters and room temperature on a given day (0-22) over the experimental period.

### 2.4 Nutrient and carbonate chemistry analyses

Dissolved nutrient concentrations were determined via spectrophotometric methods developed by Hansen and Koroleff (1999). Nitrate + nitrite ($NO_x^-$) was determined by first briefly running samples through a peristaltic pump, mixing samples with an Ammonium-chloride buffer before being passed through a cadmium reductor to reduce nitrate to nitrite. The reduced sample was mixed with sulphanilamide and N-1-naphtyl-ethylendiamine-dihydrochloride, and absorption was measured in a spectrophotometer at 542 nm. Dissolved inorganic phosphate was determined by mixing samples with ascorbic acid and a mixed reagent containing 4.5 molar $H_2SO_4$, ammonium-molybdate solution, and potassium antimony tartrate solution, forming blue heteropoly acid. The absorption of the solution was measured at 882 nm. Dissolved silicate was determined by mixing a mixed reagent containing equal amounts of molybdate solution and 3.6 molar $H_2SO_4$ with the sample, followed by ascorbic acid and oxalic acid. Sample absorbance was then measured at 810 nm. Nutrient concentrations were





calibrated with standards of known nitrate, phosphate, and silicate concentrations. The performance of the cadmium reductor and methods used for nutrient analysis were monitored by analysing the same calibration
series for each sample day and recording the absorbance and slopes of the calibration series over time. Each sample was measured in duplicate to assess technical variability between measurements. Differences were on average, 0.061, 0.001, and 0.122 µmol/L for $NO_x^-$, phosphate and silicate concentrations, respectively

The carbonate chemistry conditions were determined based on potentiometric pH and total alkalinity measurements. pH was measured daily at ~0700 hrs inside each microcosm with a Metrohm 914 pH meter and a
Metrohm Aquatrode Plus coupled glass and reference electrode, which also includes a PT1000 temperature sensor. We recorded voltage for subsequent pH calculations (see below) and temperature after observed readings had stopped drifting. This was achieved by carefully stirring the electrode for ~2-5 minutes in the upper 10 cm of the water column. pH was calibrated to the total scale ($pH_T$) using certified TRIS buffer provided by Prof. Andrew Dickson's laboratory at Scripps Institution of Oceanography as described in SOP6a by Dickson et
al. (2007). The calibration was conducted by cooling the Tris buffer to ~4°C and measuring voltage in the buffer while it was gradually warmed to 25°C. That way, we generated a temperature vs voltage correlation (26 steps along the temperature gradient), and we used the fitted equation ($R^2 = 0.999$) to obtain a reference voltage (required for the $pH_T$ calculation with equation 3 in SOP6a of Dickson et al., 2007) for every possible temperature in the microcosms. We omitted the step described by (Dickson et al. (2007) that involves the use of
AMP buffer to test for ideal Nernst behaviour of the electrode, but we note that we used a new, high-quality electrode for our measurements. Repeat measurements in buffers on different days during the experiment were within ±0.005 pH units, suggesting limited drift and comparatively high precision.

Total alkalinity (TA) was determined every fourth day with an open-cell titration following SOP3b in Dickson et al. (2007) using a Metrohm 862 Compact Titrosampler coupled with an Aquatrode Plus with PT1000
temperature sensor. Between 52-61 g of sample were added to plastic beakers (weighed using a Mettler Toledo balance with a precision of ± 0.02 mg) and acclimated to room temperature. The samples were titrated in a two-step procedure: An initial increment of 2.5 mL of ~0.05M HCl (dissolved in double deionized water enriched with 0.6 mol/kg NaCl) was added to the beaker followed by a 300 second waiting period with constant stirring. Afterward, the titration continued with additions of 0.1 mL per time step (30-60 seconds between additions
depending on drift). The titration curves were evaluated following Dickson et al. (2007) using the "calkulate" script within PyCO2sys by Humphreys et al. (2022). Certified reference material (CRM, batch 192) provided by Prof. Dickson were included in some analytical runs for accuracy control. In the runs where no CRMs were included, we included internal seawater standards (0.02% HgCl2 poisoned), which were thoroughly referenced against Dickson's CRMs. Although such procedure is clearly not recommended, this was unavoidable due to the
Coronavirus pandemic and CRM supply shortage. We note, however, that in analytical runs where both CRMs and internal standards were included, we calculated almost identical TAs, regardless of whether we used CRMs or internal standards for accuracy control. The deviation between duplicate measurements was usually below ±3 µmol/kg and rarely above ±5 µmol/kg, suggesting reasonable precision of the measurement.



Carbonate chemistry conditions were calculated from measured $pH_T$, TA, phosphate, silicate, salinity, and
temperature, with equilibrium constants recommended by Orr et al. (2015) (e.g. K1 and K2 from Lueker et al.,
2000), using the "SIR_full" function in the carbonate chemistry software 'seacarb' for R (Gattuso et al., 2021).

**2.5 Flow cytometry sampling and analyses**

Flow cytometry samples for phytoplankton (3.5 mL) and bacteria (1 mL) were collected with pipettes from the
bottles used for particulate matter sample collection (see Sect. 2.3). Care was taken to gently mix the bottles
before sub-sampling to avoid sedimentation bias. During the main phytoplankton bloom (days 4-10), we
collected additional samples in between regular sampling days to achieve daily resolution. These samples were
collected directly from the microcosms using pipettes (~5 cm below surface) after carefully stirring the
microcosms as described in Sect. 2.3. Samples were immediately fixed with 100 µL of a formalin/hexamine
mixture for phytoplankton and 20 µL glutaraldehyde for bacteria, stored at 4°C for 25 minutes, and then flash-
frozen in liquid nitrogen and stored at -80°C until analysis 1-5 weeks later. For the measurements, samples were
thawed at 37°C, then 500 µL for phytoplankton, and 30 µL for bacteria were immediately analysed with the
CYTEK Aurora flow cytometer. Phytoplankton populations were distinguished by encircling phytoplankton
populations on the cytogram plots (a.k.a "gating") based on the signal strength of the forward light scatter (FSC)
and several fluorescence colours (Fig. A2). Bacterial DNA was stained with SYBR Green I (diluted in
dimethylsulfoxide) and added to samples in a final ratio of 1:10000 (SYBR Green I:sample) prior to analysis.
This allowed us to distinguish them from other particles in the size range of bacteria (Fig. A2). Small
phytoplankton were distinguished from bacteria by excluding all particles with chlorophyll autofluorescence
from the bacteria gate.

We used the FSC signal strength to estimate how much each phytoplankton group contributed to the total
phytoplankton community during each day. For this calculation, we multiplied the abundance of each group
within a given gate by the mean FSC-Area signal strength measured for that group. Please note that "Area"
within FSC-Area refers to the integrated area below the FSC emission peak of each particle. We assume Area to
be the better metric for biomass estimates than the height of the FSC peak because elongated particles (e.g.
diatom chains) will have a more stretched-out FSC peak with a lower peak height.

**2.6 Sediment traps**

The butterfly valves at the bottom of the microcosms were initially closed so that no material could sink into the
sediment collection cups. On day 4 we opened the butterfly valves allowing water from the microcosms to enter
the collection cups. This was done to enable the sedimentation of the large aggregates which had begun to
flocculate within the microcosms (Figs. 1f, supplementary video 2). Due to the high effectiveness of our
convection mixing mechanism, which kept large aggregates in suspension, we assisted the sedimentation
process by turning off the heating and setting the room temperature to 12°C for 24 hours. On day 5 the butterfly
valves were closed, and the sediment collection cups removed, to take samples for flow cytometry and
filtrations. Fifty millilitres of water containing sedimented material was collected with a 50 mL pipette from the
base of each cup. These samples were collected in small plastic beakers and carefully homogenized before
filtering TPC/TPN and BSi samples. Filtrations followed the same procedure as described above with reduced



volumes ranging between 0.5-1 mL due to the increased concentrations of organic matter in the sediment slurry. After sampling, the cups were re-attached to the corresponding microcosm, butterfly valves opened, heating belts turned on and the room temperature returned to 7°C. The same process was repeated on days 6-7 and 8-9 with the exception that the traps were emptied entirely and cleaned on day 9 before being reattached with the

valves closed. Finally on day 12, the traps were reopened and any remaining aggregates allowed to drop out of suspension before sampling and removal of the traps from the microcosms for the remainder of the experiment. (Please note that the cleaning of collection cups during the last two samplings was conducted because the major sedimentation of organic material from the bloom was complete by day 9 and we wanted to avoid the leakage of nutrients from the collection cups back into the water column).

**2.7 Statistical analysis**

We used generalised additive mixed models (GAMMs) to assess statistically significant differences in phytoplankton growth (abundance and biomass) as well as nutrient and particulate matter concentrations over the experimental period. GAMMs were fitted using R v. 1.4.1717 and the package 'mgcv' (Wood and Wood, 2015). Prior to fitting the GAMMs, nutrient and particulate matter concentrations were log10(x) transformed

and phytoplankton count data square root transformed. Four different models were fitted to explore the potential changes in temporal trends and absolute values of each parameter as a result of alkalinity treatments (Fig. 3). All models allowed temporal trends to occur with either no difference between treatments (model 1), differences in temporal trends between treatments but no difference in absolute values (model 2), differences in absolute values between treatments but not in temporal trends (model 3), and differences in both temporal trends and

absolute values as a result of the treatments (model 4). Individual microcosms were fitted as a random intercept in each model to account for any unknown differences between the individual microcosms. In addition, heteroscedasticity and temporal autocorrelation of the residuals within models was visually assessed to ensure model assumptions were satisfied. Models were then compared by means of the Akaike Information Criterion (AIC), with lower AIC values indicating preferred models with an improved ratio between the explained

variance and number of variables. Predictor variables included in the preferred models were considered to have a statistically significant influence on the assessed parameter. Plots with fitted smoothers and corresponding confidence intervals were produced using the models with the lowest AIC value. The occurrence of significant differences between the treatments and the control could then be visually assessed by the absence of overlapping smooths and their confidence intervals between the treatments.





**3 Results**

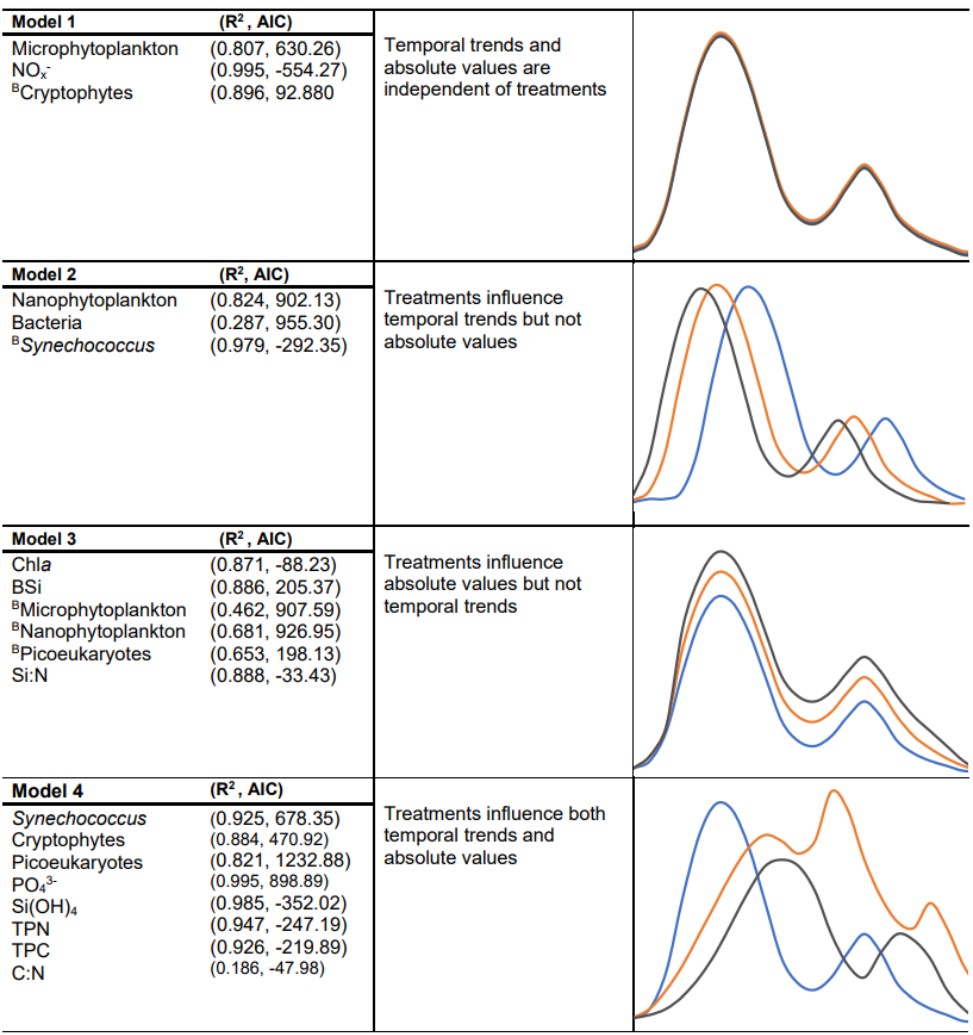

**Figure 3.** GAMM results (AIC and $R^2$) from the preferred model for each parameter with descriptive plots indicating the hypothesised smoothers for each model (phytoplankton biomass indicated with a [B] for each group). All smoothers had a $p$-value $<0.05$ indicating smoothers were significantly different from a straight line.

For a given dependent variable the model with the lowest AIC, was considered to best represent the temporal trends during the experiment and is present in the figure above.

**3.1 Carbonate chemistry and dissolved inorganic nutrients**

The addition of NaOH (for the unequilibrated treatment) and NaOH and NaHCO$_3$ (for the equilibrated treatment) resulted in an increase of total alkalinity (TA) from 2164.6 ±3.1 µmol/kg in the controls to 2660.1

±8.4 µmol/kg in the unequilibrated and 2665.2 ±2.2 µmol/kg in the equilibrated microcosms (Fig. 4a). TA remained relatively constant at these levels, apart from minor increases within the first 8 days of ~5 µmol/kg



likely due to the uptake of $NO_x^-$ during the phytoplankton bloom. The addition of $NaHCO_3$ in the equilibrated treatment increased DIC to 2406.1 ±2.1 µmol/kg, approximately 400 µmol/kg more than the control (2019.1 ±4.1) and the unequilibrated treatment (2007.9 ±9.4) (Fig. 4c). DIC decreased during the bloom with the most

pronounced decline in the control, consistent with the highest build-up of TPC (Fig. 4c, Fig. 5b). DIC gradually increased in all microcosms after bloom collapse, due to biomass respiration. $CO_2$ uptake from the atmosphere could have only had a small influence on DIC as the microcosms were tightly sealed and only opened for ~20 minutes per sampling day through a 2 cm opening. The different scenarios of alkalinity enrichment increased $pH_T$ to 8.128 ±0.009 (equilibrated) and 8.662 ±0.005 (unequilibrated) relative to 7.945 ± 0.007 in the control

(Fig. 4b). Changes in $pH_T$ reflect the phytoplankton bloom with increasing $pH_T$ until the peak of the bloom and gradually decreasing $pH_T$ thereafter. The amplitude of the $pH_T$ change during the bloom was mitigated by increased TA (Fig. 4b). However, the mitigation of the amplitude is obscured by the logarithmic nature of the pH scale, as such it is important to consider absolute changes in the free proton ($[H^+]_F$) concentration as this reflects what organisms experience (Fassbender et al., 2021), Fig. 4d). $fCO_2$ was initially 489.2 ±9.5 (control),

373.1 ±8.4 (equilibrated) and 76.6 ±0.9 µatm (unequilibrated) (Fig. 4e). The temporal trends were driven by the phytoplankton bloom and largely resembled those of $[H^+]_F$. Finally, the saturation state of the calcium carbonate ($CaCO_3$) mineral calcite ($\Omega_{calcite}$) was greatly elevated in the unequilibrated treatment with an initial value of 11.06 ± 0.03 in comparison to 2.59 ± 0.02 in the control, and 4.61 ± 0.03 in the equilibrated treatment (Fig. 4f). $\Omega_{calcite}$ increased further during the bloom but gradually declined thereafter. Inorganic precipitation of $CaCO_3$

was not observed.





**Figure 4.** Temporal variation in measured a) total alkalinity, b) $pH_T$, and calculated c) dissolved inorganic carbon, d) proton concentration on the free scale ($[H^+]_F$), e) $fCO_2$ with overlaid box plot illustrating the range of $fCO_2$ observed in the Derwent/Storm Bay area, Tasmania (42.84-43.10S, 147.46-147.31E) based on 10857 measurements between 1993-2019 (Bakker et al., 2016), f) $\Omega_{calcite}$, as well as dissolved inorganic g) nitrate + nitrite concentrations, h) phosphate concentration and i) silicate concentration within the treatment groups. Coloured shading around the respective means represents standard deviation within a treatment group.




The water enclosed within microcosms was rich in dissolved inorganic nutrients due to winter mixing. This
allowed a phytoplankton spring bloom to occur without further additions of nutrients. Initial nutrient
concentrations were 6.39 ±0.19 µmol/L for $NO_x^-$, 0.78 ±0.01 µmol/L for $PO_4^{3-}$, and 9.65 ±0.39 µmol/L for
$Si(OH)_4$. Nutrient drawdown occurred from the onset of the experiment with the most rapid drawdown
occurring between days 4-7 (Figs 4g, 4i). Statistical analysis of dissolved inorganic nutrient concentrations
revealed drawdown of $PO_4^{3-}$ and $Si(OH)_4$ to vary significantly between the control and treatments, whereas $NO_x^-$
did not (Figs. 3g, 3h, 3i). Visual inspection of the $PO_4^{3-}$ trends indicate that drawdown occurred slightly later in
the unequilibrated and equilibrated treatments when compared to the control although differences were small
(Fig. 4h). The equilibrated treatment displayed elevated $PO_4^{3-}$ values between days 10 and 14, although again
the difference was small. The drawdown of $Si(OH)_4$ was slightly delayed and considerably slower in the
unequilibrated and even more so in the equilibrated treatment (Fig. 4i). In the controls, $Si(OH)_4$ was fully
depleted on day 8 while depletion continued gradually in the equilibrated and unequilibrated treatments after the
bloom but did not show complete depletion until the end of the experiment (Fig. 4i).

### 3.2 Particulate matter and chlorophyll *a* dynamics

The drawdown of inorganic nutrients early in the experiment coincided with increasing Chl*a*, TPC, TPN, and
BSi concentrations (Figs. 5a-5d). After the peak of the phytoplankton bloom on day 6, Chl*a*, TPC, TPN, and
BSi declined relatively quickly until day 8-10 and continued to decline at a slower rate until the end of the
experiment. The alkalinity treatments had a significant influence on the temporal trends and absolute values of
TPC and TPN while they only influenced the absolute values of Chl*a* and BSi (Figs. 3, 5a-d). Visual inspection
of the data revealed similar trends in TPC and TPN, with control microcosms displaying greater concentrations
after the bloom phase for both parameters (Figs. 5b, 5c). Differences between the treatments were less apparent
for Chl*a*, with visual inspection of the trends revealing minimal differences (Fig. 5a). In contrast, BSi trends
supported the significant difference observed in the model selection process as well as the silicate trend, with
control microcosms displaying elevated levels of BSi across most of the experimental period (Fig. 5d).

### 3.2.1 Stoichiometric ratios

The molar ratio of TPC to TPN (C:N) varied both temporally and in absolute values as a result of the alkalinity
treatments. C:N declined from the initiation of the experiment until the bloom phase with the ratio of C:N then
rising rapidly in the control when compared to the alkalinity treatments which displayed a delayed increase and
lower absolute C:N value (Fig. 5e). After the bloom phase the C:N ratio was more variable between
microcosms, with the control and unequilibrated treatment having a higher C:N in comparison to the
equilibrated treatment. Similar trends between the treatments were also visible in the C:N ratio of the sediment
collection cups, with discernibly greater values in the control and unequilibrated treatment, compared to the
equilibrated treatment (Fig. 5g). The ratio of BSi to TPN (Si:N) declined rapidly from the onset of the
experiment with two small increases on day 8 and 15 (Fig. 5F). Statistical analysis of the trend revealed the
control to have a marginally higher Si:N despite the unequilibrated treatment being the greatest at the two peaks.





There was no discernible difference between treatments for Si:N ratios of organic matter from the sediment
collection cups (Fig. 5h).

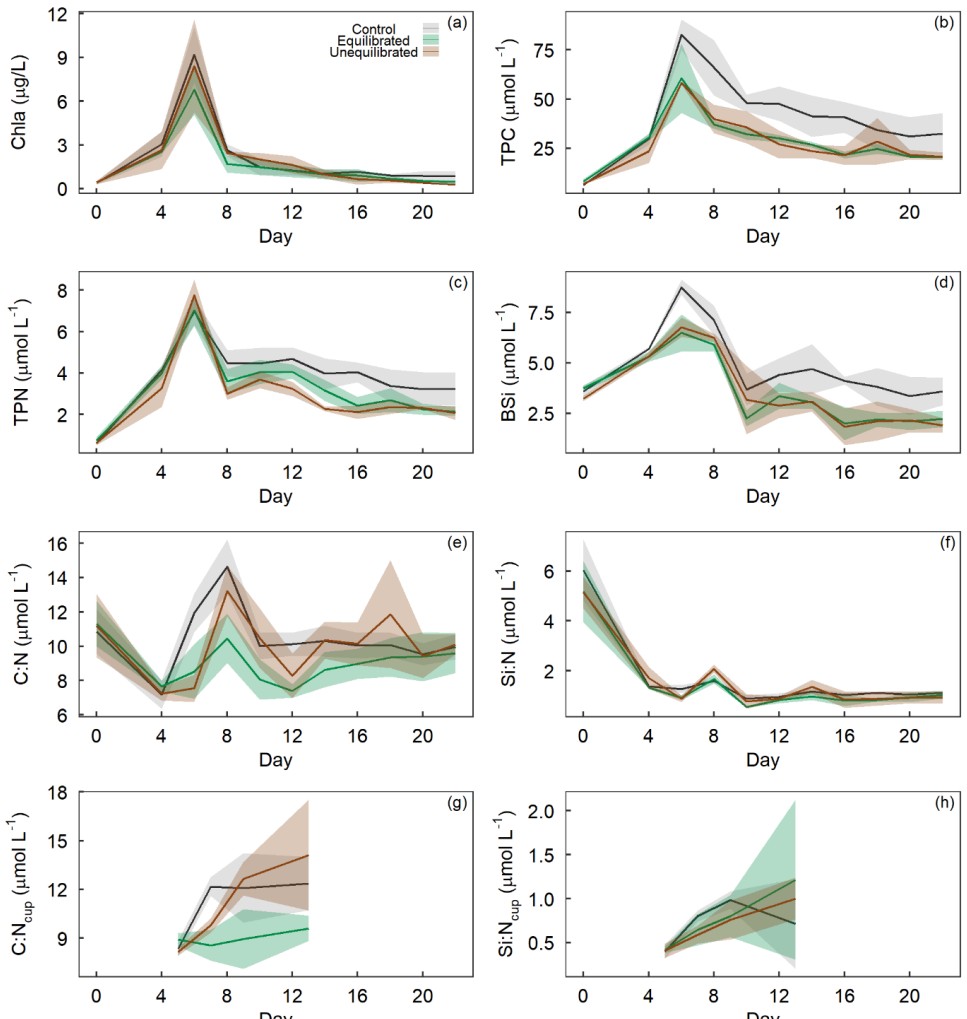

**Figure 5.** Temporal trends of A) chlorophyll *a*, B) total particulate carbon, C) total particulate nitrogen, and D)
biogenic silica concentrations, as well as molar ratios of E) TPC to TPN, and F) BSi to TPN within microcosms,
and molar ratios of G) TPC to TPN, and H) BSi to TPN within sediment collection cups, denoted by "C:N$_{cup}$" or
"SI:N$_{cup}$". Coloured shading around the respective means represents the standard deviation.

### 3.3 Changes in the phytoplankton community determined via flow cytometry

The GAMM analyses of flow cytometry count data revealed microphytoplankton to be unaffected by alkalinity
enrichment, while nanophytoplankton and bacteria showed a shift in temporal trends and *Synechococcus*,
cryptophytes, and picoeukaryotes exhibited a shift in both temporal and absolute counts (Fig. 3). In contrast,





relative biomass contributions by cryptophytes were unaffected by alkalinity treatments, whereas contributions by *Synechococcus* displayed shifts in temporal trends, and those by picoeukaryotes, nanophytoplankton, and microphytoplankton shifts in absolute biomass (Fig. 3). *Synechococcus* was initially abundant but due to their

small size their contribution to total biomass was only ~4% (Fig. 6a, 6b). *Synechococcus* abundance declined from the start of the experiment in both alkalinity treatments, while the decline occurred two days later in the control (Fig. 6a). There were also significant temporal differences between treatments in *Synechococcus* biomass, with an earlier decline in the equilibrated treatment followed by the control and then unequilibrated treatment (Figs. 3, 6b). After day 8, *Synechococcus* abundance remained close to the detection limit and

provided minimal contribution to the plankton community biomass thereafter (Figs 6a, 6b). Picoeukaryote abundance and biomass showed little variation between the control and equilibrated treatment throughout the experiment but was significantly smaller and slightly delayed in the unequilibrated treatment during the bloom (Fig. 6c). This trend was reflected in the biomass contribution of picoeukaryotes, which was notably lower in the unequilibrated treatment during the bloom (Fig. 6d). Cryptophytes contributed up to 20% to the total

plankton biomass with no temporal or absolute difference between treatments (Fig. 3). Cryptophyte abundance was significantly elevated and peaked earlier in the control compared to the two alkalinity treatments. After the bloom, cryptophyte abundance declined close to or below the detection limit in all treatments and did not contribute significantly to total phytoplankton biomass thereafter (Figs. 6e, 6f). Nanophytoplankton abundance increased during the bloom phase of the experiment but there was no significant difference observed between

the treatments. However, during the post bloom phase abundances were significantly elevated in the unequilibrated treatment (Fig. 6g). The nanophytoplankton group initially contributed ~60% to phytoplankton biomass, with marginally greater biomass in the unequilibrated treatment in comparison to the equilibrated treatment over the extent of the experimental period (Fig. 6h). Microphytoplankton abundances increased during the bloom and peaked on day 6 but as analysis revealed model 1 to be the preferred model, we conclude that

there were no statistically significant differences between the treatments (Fig. 3, 6i). However, there was a significant trend in microphytoplankton contribution to total biomass, with a peak of ~35% during the bloom phase before dropping to ~1-25% for the remainder of the experiment (Fig. 5j). Microphytoplankton contributed marginally but significantly more biomass in the control microcosms during the last six days of the study (Fig. 5j). Finally, bacteria showed variations in temporal trends as a result of the treatments with a greater abundance

in high alkalinity treatments during the phytoplankton bloom and more constant abundances throughout the experiment, whereas abundances in the control were low during the bloom but increased rapidly thereafter (Fig. 5k).



**Figure 6.** Temporal trends of phytoplankton group abundance (left column) and percent biomass contribution (right column) determined by flow cytometry. Group names provided in the top right of each plot. Coloured shading around the respective means represents standard deviation within a treatment group.





## 4 Discussion

Alkalinity had a noticeable influence on the characteristics of the phytoplankton bloom and associated succession of the phytoplankton community. However, finding unequivocal explanations for how alkalinity
altered succession patterns is very difficult in this form of community experiment due to the numerous degrees of freedom in complex food webs. Therefore, we use the discussion henceforth to present potential explanations, which we believe to be particularly plausible while emphasizing that none of these can be exclusively proven or excluded. This leads to many speculations with regards to data interpretation as the reader will likely notice in the text below. However, our observations are still highly valuable as they reveal important patterns and any
strong effects of alkalinity on components of the phytoplankton community that can then be investigated in more targeted future studies.

### 4.1 Alkalinity effects on chlorophyll *a*, carbon, nitrogen, and silicon dynamics

#### 4.1.1 Build-up of chlorophyll a during the phytoplankton bloom

A significant difference in chlorophyll *a* of ~3 µg/L was observed between the control and equilibrated
treatments during the peak of the phytoplankton bloom, while no significant differences were observed between the control and unequilibrated treatment. The lower peak chlorophyll *a* in the equilibrated treatment was unexpected as carbonate chemistry parameters believed to predominantly drive phytoplankton growth ($CO_2$ and $H^+$; Paul and Bach 2020) were relatively similar to the control and within natural ranges (Fig. 4d, e). We suspect the low peak in chlorophyll *a* concentration may be due to differences in the predominant species driving
chlorophyll *a* build-up. This is supported by careful inspection of the raw flow cytometry data where we noticed that different types of phytoplankton occurred within the flow cytometry gate denoted as nanophytoplankton (Fig. A3). The majority of the population was closer to the upper edge of the nanophytoplankton gate in the equilibrated treatment in comparison to the control on day 6. Although speculative, lower concentrations of chlorophyll *a* could also be due to increased grazing in the equilibrated treatment. However, the influence of
grazing was not assessed in this study.

In contrast, and even more unexpected, there was no significant difference in peak chlorophyll *a* between the control/equilibrated and the unequilibrated treatment. The $fCO_2$ was as low as ~70 µatm in the unequilibrated treatment, which is substantially lower than what is encountered by phytoplankton in coastal Tasmania over the course of a season (Fig. 4e, A4, see also: C. Pardo et al., 2019). Previous studies have revealed that growth rates
of phytoplankton are relatively unaffected by low $CO_2$, as long as $CO_2$ concentrations are only mildly reduced (Riebesell et al., 1993, Wolf-Gladrow et al. 1999). However, rapid declines in growth were frequently observed once $CO_2$ concentrations fell below species-specific thresholds, with such thresholds usually being well above 70 µatm (Riebesell et al., 1993; Chen et al., 1994; Hinga, 2002; Berge et al., 2010; Paul and Bach, 2020). Based on these studies, we expected a delay in the peak of the phytoplankton bloom and/or reduced bloom intensity.
The fact that neither of these occurred suggests that the phytoplankton species growing during the bloom were either (1) unaffected by (i.e. well-adapted to) low $CO_2$ or (2) that certain species within the community were adapted to low $CO_2$ and could compensate for less well-adapted species. While our data does not provide a definitive answer to this, there are two arguments that favour the second explanation. First, BSi build-up and corresponding $Si(OH)_4$ drawdown strongly suggest that the alkalinity treatments affected the diatom community



during the bloom. Second, there were significant differences in picoeukaryote and cryptophyte abundances during the bloom, with lower abundance and contribution to biomass in the unequilibrated treatment (see Sect. 4.3 for further discussion on picoeukaryote responses). Together, these observations suggest that the addition of alkalinity without immediate $CO_2$ equilibration with the atmosphere may have less of an impact on phytoplankton bloom dynamics than previously thought. However, phytoplankton species composition may still

be affected, with implications for energy transfer to higher trophic levels and biogeochemical fluxes, both of which are strongly dependent on phytoplankton species composition (Mallin and Paerl, 1994; Wassmànn, 1997).

### 4.1.2 Carbon and nitrogen dynamics during and after the bloom

TPC, TPN and the C:N ratio were all significantly greater in the control compared to the high alkalinity

treatments during the phytoplankton bloom (days 4-8). In contrast, minor differences were observed between the two alkalinity treatments during this period. Previous experiments have shown that carbonate chemistry conditions can affect the build-up and stoichiometric relationship of organic carbon and nitrogen, but the effect is highly variable and dependent on the composition of the plankton community (Taucher et al., 2020). The key outcome reported by Taucher et al. (2020) was that heterotrophic processes seem to have an important influence

on C:N stoichiometries. Consistent with their observation, we observed significant increases in TPC and C:N in the control during the bloom, while bacterial abundances remained relatively low (compare Figs. 5e, 6k). In contrast, bacterial abundances were significantly higher in the alkalinity treatments, indicative of higher respiratory activity, which may have limited the build-up of TPC (Figs. 5e, 6k). Furthermore, differences in diatom growth and/or community composition between the control and the alkalinity treatments (discussed in

section 4.1.3) can also offer a direct explanation for the differences in TPC build-up and C:N ratios observed during the bloom. Diatoms often dominate phytoplankton blooms where they exude DOC which partially aggregates to form 'transparent exopolymer particles (TEP)' (Passow, 2002). TEP have high C:N ratios which commonly exceed the Redfield ratio (Engel and Passow, 2001) and would be part of the TPC pool measured in our study. The production of TEP has been found to vary significantly between diatom species with a laboratory

study revealing four species to produce significantly different concentrations of TEP per cell volume (Fukao et al., 2010; Passow, 2002). As such it is plausible that alkalinity treatments altered the abundance and/or composition of the diatom community (see Sect. 4.1.3.) leading to less TEP, measurable as higher TPC build-up and C:N.

Diatoms are between a few micrometres to a few millimetres in size (Armbrust, 2009). The largest diatom cells

in our experiment were roughly 50 µm so all diatom cells are most likely found in the nano- and microphytoplankton groups in flow cytometry data. Although not statistically significant, visual inspection of microphytoplankton abundance during the peak of the bloom (day 6) revealed greater abundances in the unequilibrated treatment followed by the control and then equilibrated treatment. This indicates differences in the phytoplankton communities between the treatments and the control with potential influence on TPC build-up

and C:N ratios. In addition, significant differences in the build-up of BSi and drawdown of $Si(OH)_4$ between the control and treatments also strongly suggest that the alkalinity treatments caused variation in the diatom communities.



In summary, the evidence provided herein suggests that the altered carbonate chemistry conditions due to elevated alkalinity caused changes in the autotrophic and heterotrophic communities which collectively altered
TPC build-up and C:N ratios. Accordingly, anthropogenic increase of ocean alkalinity may have the capacity to influence ecological processes with implications for biogeochemical processes. Crucial next steps are to confirm such impacts in community studies, other environments and to reveal the underlying mechanism(s) responsible for triggering the observed community changes in response to alkalinity additions.

### 4.1.3 Biogenic silica and dissolved inorganic silicate drawdown

Scanning electron microscopy investigations of samples taken before, during, and after the phytoplankton bloom revealed that diatoms were the only silicifiers detected in the plankton community. Therefore, the drawdown of $Si(OH)_4$ and build-up of BSi within microcosms can be attributed to the diatom community. BSi increased during the peak bloom before declining and remaining rather constant from day ~12 onwards with significantly higher concentrations in the control than in the alkalinity treatments (Fig. 5d) The greater concentration of BSi
in the control is consistent with a more complete drawdown in $Si(OH)_4$ (Figs. 4i, 5d). There was no significant difference observed in the build-up of BSi between the two alkalinity treatments even though the drawdown of $Si(OH)_4$ was significantly greater in the unequilibrated treatment. There are two Si pools that were not quantified in our study where the additional Si consumed in the unequilibrated treatment could have gone. These are (i) the walls of the microcosms where benthic diatoms may have grown and consumed Si, or (ii) the
sediment traps where relatively more BSi from sinking diatoms may have been collected (please note that we quantified elemental ratios of sinking organic matter collected in the sediment traps but not total mass flux as this requires sampling of all collected material for which we did not have the capacity).

The significant and pronounced differences in $Si(OH)_4$ drawdown/BSi build-up between the control and the alkalinity treatments are arguably one of the most striking observations in this experiment. It suggests that
alkalinity enhancements and associated changes in carbonate chemistry can have considerable effects on diatom communities. Carbonate chemistry changes invoked by simulated ocean acidification have been shown to have a significant influence on BSi content, silicate metabolism, growth, and diatom silicification (Milligan et al., 2004; Hervé et al., 2012; Petrou et al., 2019) albeit the sign and magnitude of diatom responses were species-specific and dependent on the communities investigated (Bach and Taucher, 2019; Petrou et al., 2019; Pedersen
and Hansen, 2003). To the best of our knowledge, there is currently no established mechanistic framework that can explain the variable responses of diatoms to carbonate chemistry, although useful concepts exist that link the carbonate chemistry sensitivity to diatom size (Flynn et al., 2012; Wolf-Gladrow and Riebesell, 1997; Wu et al., 2014). The observation is also remarkable because the differences in BSi occur between the control and both alkalinity treatments even though differences in 'physiologically-relevant carbonate chemistry parameters' (i.e.
mainly $CO_2$ and $[H^+]_F$; Shi et al., 2019) are much larger between the equilibrated and unequilibrated alkalinity treatments (Figs. 4d, 4e). This suggests that (i) an unexpected factor in the carbonate chemistry drove the diatom response or (ii) that the carbonate chemistry effect on diatoms was indirect, e.g. transmitted through altered grazing pressure. The second scenario could for example be caused by the additions of acid and base in the treatments, which may have harmed the grazers and affected the grazing pressure. Either of these (or other)
physiological/ecological explanations for the alkalinity effects on BSi build-up/$Si(OH)_4$ drawdown should be visible as a change in the diatom abundance and/or community composition. For example, there could be a shift



in the diatom community towards smaller, less silicon-containing species and/or a higher fraction of non-silicifying species. To explore this possibility, we analysed the diatom community at peak bloom (day 6) via scanning electron microscopy. However, there were no clear differences in diatom community biovolume
between the control and alkalinity treatments on day 6 (Fig. A5). Thus, although we suspect that shifts within the diatom community were responsible for the observed differences in silicon dynamics, we are currently unable to provide a definitive mechanism for these observations.

**4.2 Effect of alkalinity on the phytoplankton community determined via flow cytometry**

Alkalinity treatments were found to significantly influence the abundance and biomass of five out of the six
phytoplankton groups assessed via flow cytometry and analysed using GAMMs (Fig. 3). The majority of the detected differences were in absolute values during the peak bloom and small temporal shifts between treatments.

Comparatively pronounced differences between treatments and the control were identified within the groups
*Synechococcus*, cryptophytes, and picoeukaryotes where alkalinity treatments negatively influenced abundance during the bloom phase and/or delayed the peak bloom. The unequilibrated treatment had the greatest influence on these groups, suggesting that the significantly lower concentration of $CO_2$ and/or increased pH negatively affected these groups. Previous micro- and mesocosm research on ocean acidification have found variable responses of *Synechococcus* and cryptophytes, indicating that their responses to carbonate chemistry may be (i)
population-specific, thus varying between experiments or (ii) transmitted indirectly through food web interactions, which also vary across experimental communities (Sala et al., 2016; Schulz et al., 2017; Bach et al., 2017).

The response of picoeukaryotes to ocean acidification (i.e. increasing CO2, declining pH) has been remarkably
consistent through experiments in various climatic and experimental settings (Thomson et al., 2016; Maugendre et al., 2015; Sala et al., 2016; Schulz et al., 2017; Davidson et al., 2016; Hoppe et al., 2018; Newbold et al., 2012; Schaum et al., 2012; White et al., 2020). Our results are consistent with these findings as we reveal the opposite trend occurred when carbonate chemistry changes were reversed, i.e. when we decrease $CO_2$ and increase pH, we observe a reduction in picoeukaryote abundance. This is illustrated by the equilibrated
treatment where relatively small differences in $CO_2$ and pH result in little to no differences in picoeukaryote abundance, whereas large differences between the control and unequilibrated treatment had a pronounced effect on picoeukaryote abundance (Fig. 6c). It has been speculated that the influence of $CO_2$ on picoeukaryotes is due to their increased reliance on diffusive $CO_2$ entry in comparison to other functional groups which rely more heavily on carbon concentrating mechanisms (CCMs) and the substantially larger $HCO_3^-$ pool (Crawfurd et al.,
2016; Meakin and Wyman, 2011; Engel et al., 2008). The operation of CCMs is energetically costly, however larger cells have been revealed to be more efficient at transporting carbon using CCMs with a reduction in $CO_2$ leakage as a function of size (Engel et al., 2008; Malerba et al., 2021). Within this framework, smaller cells such as picoeukaryotes would be at a disadvantage at lower $CO_2$ concentrations in comparison to larger cells (Malerba et al., 2021: Meakin and Wyman 2011). Our results support this as picoeukaryotes were apparently



more sensitive to low $CO_2$ or high pH than the larger phytoplankton groups such as microphytoplankton
(discussed below).

Differences between the treatments were less apparent for the nanophytoplankton group, with no differences
during the bloom phase and slightly greater abundance during the post bloom phase for the unequilibrated
treatment. The nanophytoplankton group contributed the largest proportion to total biomass of all the assessed
groups, increasing from 55-65% at the initiation of the experiment up to 95% at the end. The nanophytoplankton
cluster in the flow cytometer is usually variable across or within treatments as there are many species in this
approximate size range that could be captured. It is therefore possible, if not likely, that there was a succession
towards different nanophytoplankton species between the control and treatments, which may explain different
succession patterns. Treatment-specific differences in nanophytoplankton abundances are usually hard to
interpret as it is mostly unclear what species are contributing to the cluster and what physiological/ecological
responses to perturbation we can expect.

The microphytoplankton group did not display statistically significant differences in absolute abundances or
temporal shifts for cell counts. However, as discussed in section 4.1.2, we argue that there may have been higher
microphytoplankton abundances in the unequilibrated treatment during the peak of the phytoplankton bloom
(Fig. 6i), but this was too short to be detected as a significant difference in the statistical analysis. The absence
of a negative effect of low-$CO_2$/high-pH in the unequilibrated treatment was surprising as theory predicts more
pronounced constraints on diffusive $CO_2$ uptake of larger phytoplankton species (Wolf-Gladrow and Riebesell,
1997; Flynn et al., 2012). Our experimental approach does not reveal how this absence of an effect could be
explained. As argued in sections 4.1.2 and 4.1.3 we speculate that the most likely explanation is a shift in the
species composition where species that are more capable at low-$CO_2$/high-pH conditions may have compensated
for those with reduced capacity. This important observation warrants further investigation.

**4.3 Implications of the environmental assessment of ocean alkalinity enhancement**

The amount of alkalinity added in our experiment increases the capacity of seawater to store atmospheric $CO_2$
by 21%. It is crucial to understand that this is a massive enhancement of the inorganic carbon sink of seawater.
For example, 21% of all DIC in the ocean equals ~8000 GtC, >10 times more than all carbon emissions since
1750 (Friedlingstein et al., 2019). The inadvertent effect of a 21% sink enhancement on the phytoplankton
community seems justifiable in our experiments in relation to the substantial benefits such permanent (>>1000
years) $CO_2$ storage would have for the climate. Other marine $CO_2$ removal methods such as Ocean Iron
Fertilisation are likely associated with at least equally pronounced perturbations of the phytoplankton
community (Quéguiner, 2013), for the benefit of an approximately 1% non-permanent (<100 years)
enhancement of the marine carbon sink observed during mesoscale iron fertilization experiments in the Southern
Ocean (Bakker et al., 2005).


One particularly interesting observation was that the unequilibrated alkalinity treatment was not noticeably more
affected by the perturbation than the equilibrated treatment (Figs. 5, 6), despite substantially larger differences
in carbonate chemistry relative to the control (Fig. 4). This is of significant importance as equilibrated alkalinity



additions will likely be associated with additional costs, due to engineering efforts and energetic requirements of

equilibrating systems (e.g. $CO_2$ bubbling and associated pumping). However, the release of alkalinity into the

marine environment without a controlled influx of atmospheric $CO_2$ leads to verification challenges as it

remains unclear where and when the $CO_2$ influx will occur (Orr and Sarmiento, 1992; Gnanadesikan and

Marinov, 2008; Bach et al., 2021). Verification is important to refinance and incentivize $CO_2$ removal efforts

(Hepburn et al., 2019; Rickels et al., 2021). Thus, if not for environmental reasons, an engineered and controlled

influx of atmospheric $CO_2$ after alkalinity additions as tested in the 'equilibrated' treatment may still be

important for economic reasons.

One limitation of our experimental microcosm setup was the consistently high alkalinity ($+498 \pm 5.2$ µmol/kg) in

the treatments for the entire 22-day experiment. In real-world OAE applications, alkalinity-enriched seawater

from point-sources (e.g. electrochemical facilities (de Lannoy et al., 2018)) or mineral powder-enriched surface

ocean areas (Renforth and Henderson, 2017) will be diluted over time with surrounding seawater of lower

alkalinity. The degree of dilution with unperturbed water is site specific and depends on the type of application

(e.g. more dilution for a small point source in a system with high mixing rates). It can be expected that the

dilution of alkalinity-enriched seawater would weaken the impact of alkalinity on the plankton community

because of decreasing changes in carbonate chemistry relative to the non-perturbed state. Thus, our

experimental setup simulated a relatively high intensity of perturbation as any impact mitigation through

dilution is excluded.

OAE can be achieved through a variety of approaches, ranging from distributing pulverized minerals onto the

sea surface to splitting water into acid and base using electrochemistry (Renforth and Henderson, 2017). All

methods seek to increase surface ocean alkalinity, but the by-products generated in the various processes are

highly variable. In this study, we utilised laboratory grade NaOH to increase the alkalinity of microcosms, a

perturbation scenario representative of OAE via the electrochemical splitting of water (de Lannoy et al., 2018).

Here, no other chemicals than strong acid (HCl) and base (NaOH) are generated and only the base is released

into the surface ocean (de Lannoy et al., 2018; Tyka et al., 2022). OAE approaches associated with the release

of other bioactive components such as trace metals could have more substantial effects on the plankton

community. We emphasise this aspect to stress that our observations of relatively moderate impacts of

'equilibrated' and 'unequilibrated' alkalinity perturbations cannot be generalised for all OAE approaches. From

this perspective, our simulated perturbation arguably tested a 'mild' version of OAE. The environmental

assessment of OAE needs to remain in close contact with geochemical research in order to anticipate which

OAE approaches have the greatest chance for upscaling. This will allow for a targeted assessment of the

perturbations associated with the OAE approaches most likely to be implemented in the future.

**5. Conclusion and outlook**

This study is the first study to report on the effects of OAE on a coastal plankton community. Our key findings

are:



1)  Two different scenarios of alkalinity enhancement ('CO$_2$-equilibrated' with the atmosphere and 'unequilibrated') had a significant influence on the succession of the phytoplankton community and heterotrophic bacteria.


2)  There were pronounced effects of alkalinity enhancement on diatoms even though dissolved Si concentrations were not manipulated in this study.

3)  Consistent with previous research on ocean acidification we found that low-CO$_2$/high-pH conditions are detrimental for picoeukaryote phytoplankton.

4)  Surprisingly, the unequilibrated alkalinity treatment did not have a noticeably greater effect on the phytoplankton community than the equilibrated treatment, despite much larger changes in


physiologically important carbonate chemistry parameters.

Altogether our findings suggest that sudden increases of alkalinity leave a noticeable imprint on the succession of the phytoplankton community. However, the environmental effects investigated here appeared to be moderate when compared to the enormous climatic benefit of increasing the inorganic carbon sink of seawater by 21%.

It is generally problematic to quantify changes in plankton communities as positive or negative as this depends on the perspective. More than two decades of ocean acidification research have shown that there will be winners and losers in plankton communities when carbonate chemistry is perturbed (Schulz et al., 2017; Alvarez-Fernandez et al., 2018; Taucher et al., 2020). These shifts were often perceived as negative (Falkenberg et al., 2020; le Quesne et al., 2012; Doney et al., 2020) but occasionally also as positive (Sswat et al., 2018; le Quesne

et al., 2012). Mixed (or perspective-dependent) outcomes can also be expected for the assessment of OAE. From a human perspective, plankton community shifts affecting trophic transfer and ultimately fish production are comparatively easy to quantify as positive or negative. Our dataset did not provide insights on this aspect as we focussed only on the lowest trophic level. It is possible that the seemingly moderate effects of alkalinity observed at the lowest trophic level could have been amplified in higher trophic levels. Future studies should

aim for a comprehensive assessment of higher trophic levels to better understand how lower trophic level change affects upper trophic levels and also to reveal potential top-down effects of OAE. Furthermore, other pelagic and benthic ecosystems, from arctic to tropical, need to be investigated to gather a reliable and comprehensive assessment of OAE effects on marine ecosystems. This study can therefore only be seen as a small first step.





**6. Appendix A**

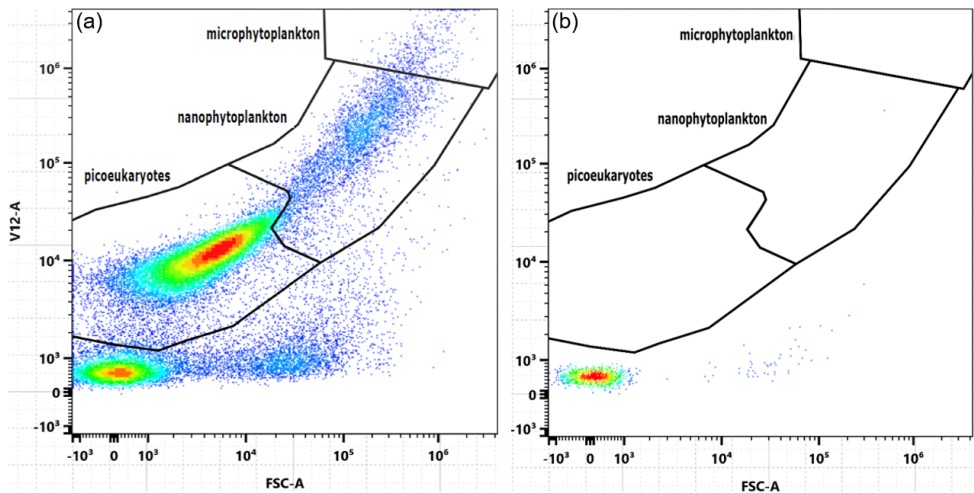

**Figure A1.** Cytograms used to determine the size of particles filtered by QMA filters used in TPC and TPN analysis. Plot (a) depicts a water sample filtered through a QMA filter (2.2 µm) and plot (b) an unfiltered sample. Both plots were produced using the same sample from microcosm M4 on day 6.

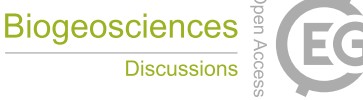




**Figure A2.** Gating strategy when analysing data via flow cytometry. Plot (a) illustrates the intensity of fluorescence for each channel in the total sample. Plots (b)-(d) show gates for picoeukaryotes, nanophytoplankton, microphytoplankton, Synechococcus, and cryptophytes in microcosm M3 on day 5. Plot (e) shows the gate for bacteria in microcosm M4 on day 12.



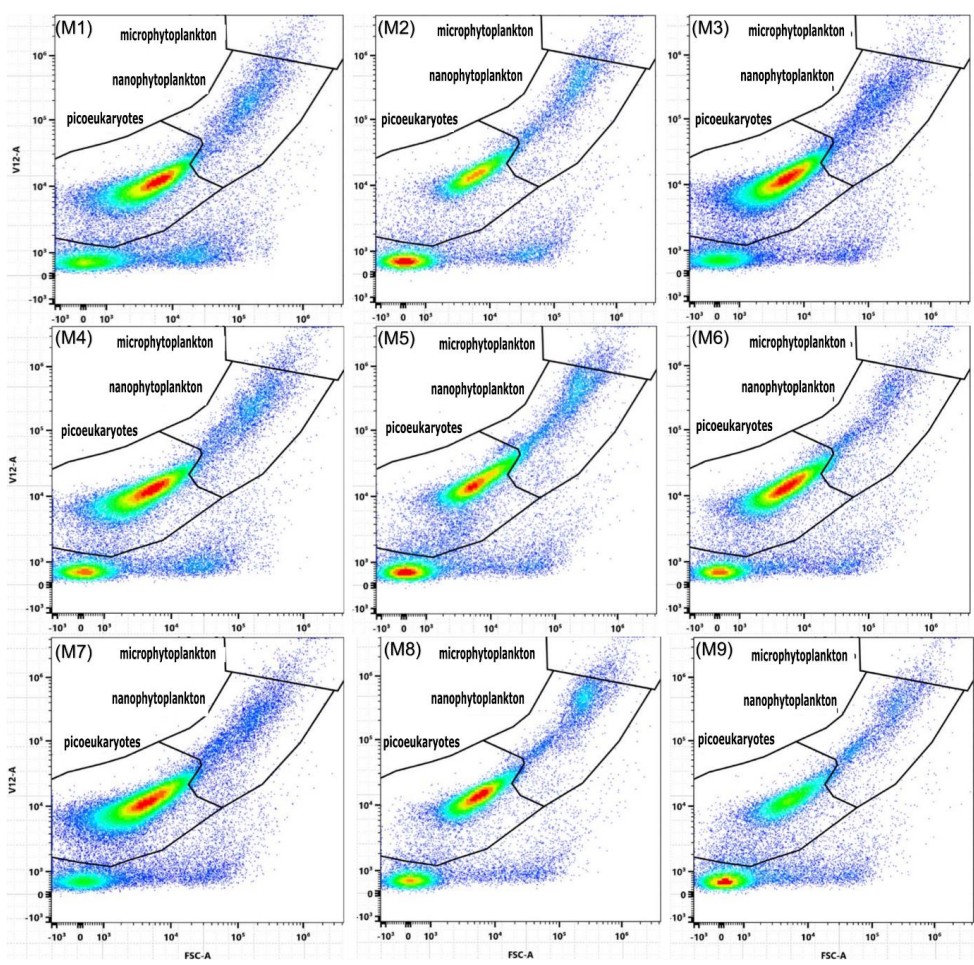

**Figure A3** Cytograms depicting differences within gates, between treatments. Plots are labelled to corresponding microcosms so that M1, M4, M7 represent the unperturbed control, M2, M5, M8 represent the unequilibrated treatment and M3, M6, M9 represent the equilibrated treatment. All plots are from samples taken during the peak bloom on day 6.



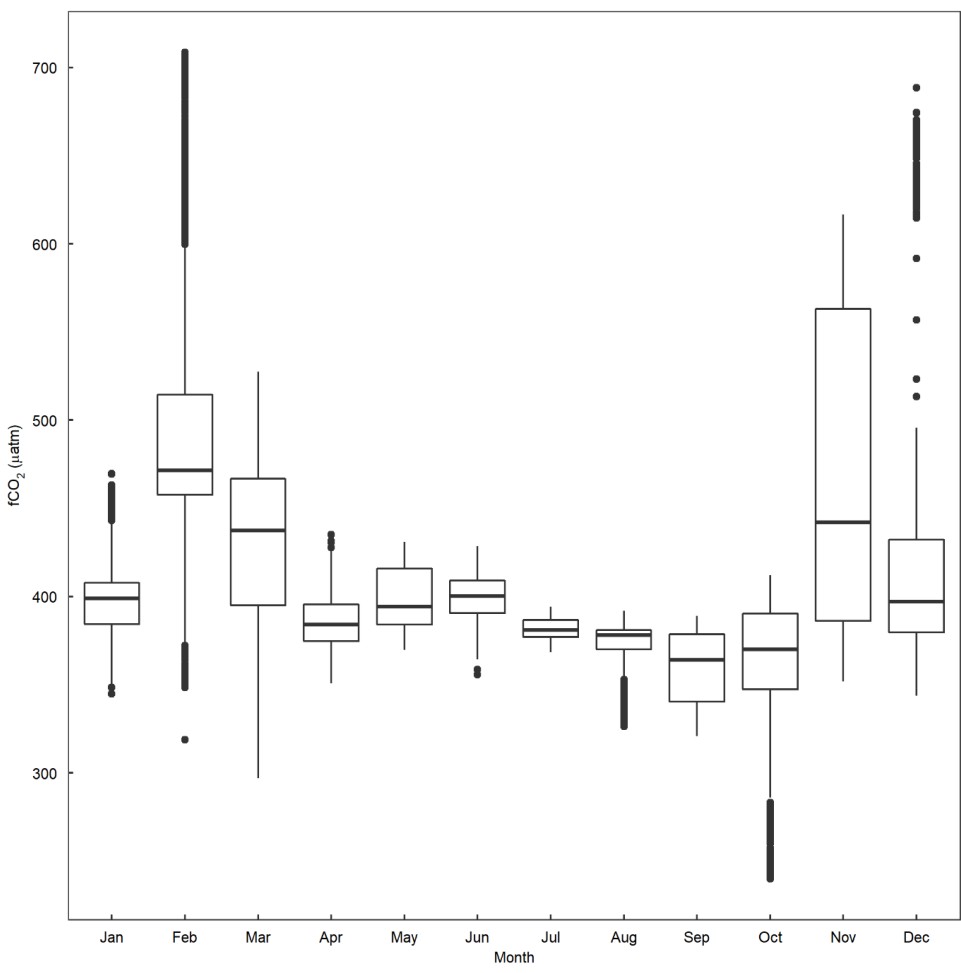

**Figure A4** Boxplot depicting seasonal values of fCO-$_2$ recorded between 1993-2019 at Storm Bay (43.1 - 42.8442S, 147.307 - 147.46E), Tasmania (Bakker et al., 2016).






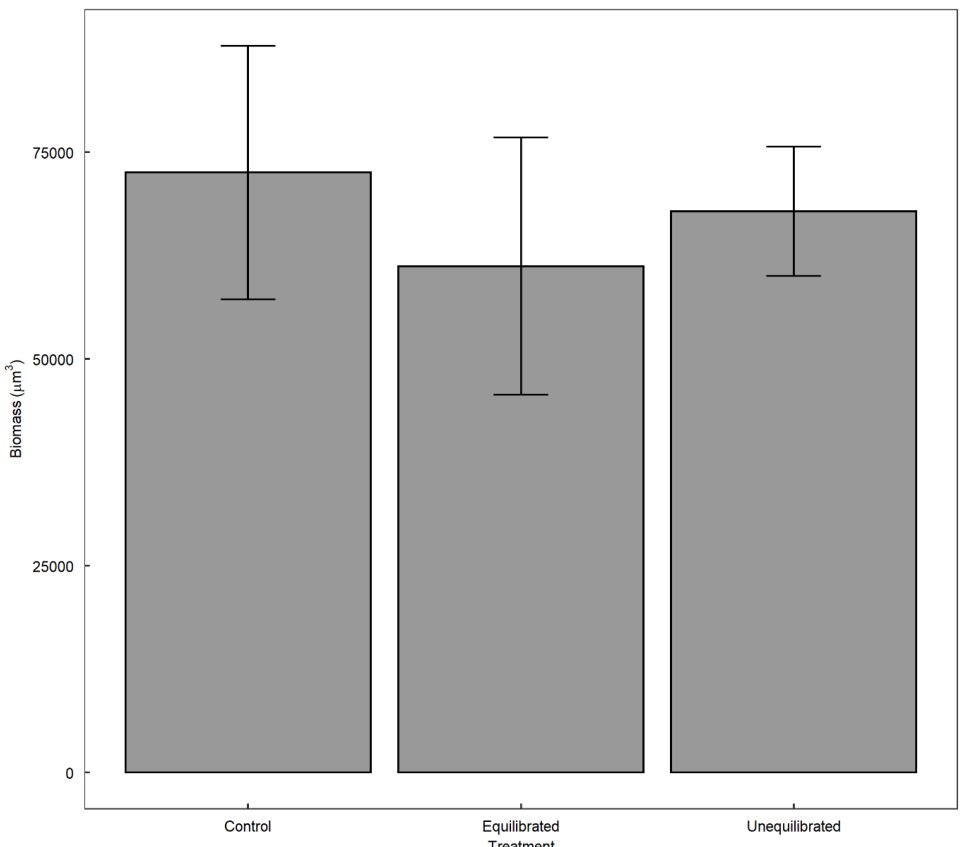

**Figure A5** Average diatom biomass during the peak bloom (day 6) within treatments determined via
microscopy.

**7. Data availability**

Data is available from the Institute for Marine and Antarctic Studies (IMAS) data catalogue, University of
Tasmania (UTAS). Ferderer, A. (2021). *Assessing the influence of ocean alkalinity enhancement on a coastal
phytoplankton community - manuscript data* [Data set]. Institute for Marine and Antarctic Studies (IMAS),
University of Tasmania (UTAS). https://doi.org/10.25959/8PEA-SW88

**8. Video supplement**

Video supplement 1. Contains a time-lapse of the convective mixing test described in sect. 2.1, taken on 3
August 2021. The video can be accessed online at: https://doi.org/10.5446/55861.
Video supplement 2. Contains a time-lapse depicting aggregate formation and suspension within a microcosm as
a result of convective mixing. Taken on 15 August 2021. The video can be accessed online at:
https://doi.org/10.5446/55860.



### 9. Author contributions

LTB designed the experiment. AF was responsible for the investigation with the help of LTB and FK. AF was
also responsible for the data curation, formal analysis, and writing. LTB and FK supervised data collection. AF
wrote the manuscript with contributions from LTB, FK, ZC and KS.

### 10. Competing interests

The authors declare that they have no conflict of interest.

### 11. Acknowledgements

We would like to thank: Sandrin Feig, Thomas Rodemann, and Terry Pinfold for support on scanning electron
microscopy, particulate organic matter, and flow cytometry measurements. This research was funded by a
Future Fellowship (FT200100846) by the Australian Research Council awarded to LTB. This research was also
conducted while AF was in receipt of an Australian Government Research Training Program (RTP) scholarship.

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
