# Peer review of "Assessing the influence of ocean alkalinity enhancement on a coastal phytoplankton community"

_Biogeosciences, 2022_

## Author Response (AR1)

Dear referees,

Thank you for your comments on our manuscript. We appreciate the time and effort you have dedicated to providing valuable feedback on our manuscript. Here are our point-by-point responses to your comments.

**Comment 1:** The first reservation I have is the duration of the reported experiment, which appears to have been too long given the size of the bottle. The experiment lasted for more than 20 days. Did the authors consider possible biases in species composition because of the bottom effect? I am not convinced that it is possible to interpret with confidence biology data obtained after 10 days of experimentation, and any conclusion drawn about changes in phytoplankton communities based on such data may well be erroneous. The bottle effect is well known and can seriously bias results. The authors should provide convincing evidence that this effect has not influenced their results.

**Response 1:** The reviewer addresses a fundamental limitation of all experimental ecological studies with enclosures. Bottle effects always affect the outcome in experimental ecological studies, regardless of the size of the container (Carpenter, 1996; Duarte et al., 1997). Importantly, however, there is no evidence to suggest that a specific duration (e.g. 10 days as suggested here) is an ideal timeframe to minimise bottle effects (Duarte et al., 1997). From our experience, even very large and sophisticated floating mesocosms enclosing ~50,000 L of natural seawater induce 'bottle effects' right from the beginning (day 1) as evidenced from much stronger fluctuations in phytoplankton communities inside the mesocosms than in the surrounding seawater (see figure 3 in Bach et al., 2019). Thus, we disagree that 20 days is too long as there is no basis for such an argument. Twenty days was chosen as we were able to cover a nutrient-induced phytoplankton bloom and the subsequent post-bloom phase with sufficient data points to obtain meaningful insights. With regards to biases in the community due to bottle effects, as noted above, this is a general problem and we therefore agree that bottle effects can drive changes in community composition (as is discussed in length in Bach and Taucher, 2019). However, these bottle effects occur in all micro and mesocosms. Thus, we implicitly account for them so that the comparison between control and treatment are still valid. Or in other words, while there may be deviations from natural plankton successions due to bottle effects, the conclusions drawn from the comparisons between the control and treatments are still technically sound. We will add a clarification to the manuscript to account for the reviewer's valid concern: (Lines 126-132) "Nevertheless, despite some potential advantages, we acknowledge and are fully aware that our microcosm setup cannot reproduce the full physical (or chemical/biological) complexity of nature (Carpenter, 1996). Enclosures of any type will very likely induce so-called bottle effects (Bach and Taucher, 2019), which can alter the observed community succession and therefore affect the transferability of the outcome to natural (non-enclosed) communities (Carpenter, 1996). While this is a general limitation of these kind of experimental studies, we stress that bottle effects would occur in all replicates so that the comparison between control and treatments (as done in our study) is valid."

**Comment 2**: The second reservation I have is the validity of phytoplankton composition data obtained after the complete depletion of nutrients. Nearly all the N and P was depleted at day 6 (Si was depleted at day 8), but the authors reported phytoplankton composition changes for 20 days after nutrients had completely run out. I think the effects of alkalinity increase on phytoplankton composition should be evaluated under the ample nutrient

conditions. Once the nutrients were depleted the phytoplankton would have been affected by both alkalinity change and nutrient constraints. These two variables may have in combination contributed to phytoplankton composition change. The authors should explain how they differentiated the effect of alkalinity change from the effects of nutrient constraints.

**Response 2:** The goal of this experiment was to observe changes in the phytoplankton community over various stages of a bloom, including when inorganic nutrients were depleted, in excess and declining. We emphasise that phytoplankton are nutrient-limited for most of the seasonal cycle. Thus, assessing the effects of OAE on communities under nutrient-depleted conditions is at least equally as important as nutrient-replete conditions. It is important to keep in mind that we initially enclosed the same water with similar nutrient concentrations (with random, non-systematic variation). We agree that differences in plankton communities later in the experiment may have been induced by the different (residual) nutrient concentrations after the bloom. However, these different concentrations were then a result of the treatments, established on the first day of the study. For example, the equilibrated OAE treatment had a pronounced effect on silicate drawdown. It is possible that this difference in silicate concentration had a downstream effect on the community in the post-bloom phase. However, the silicate difference is a result of the treatment and thus ultimately an important effect of OAE, transmitted by changes in nutrient drawdown. To account for this valid concern raised by the reviewer we will add the following clarification (lines: 585-590) "The aim of this experiment was to assess the influence of alkalinity enhancement on the various stages of a spring bloom. This included periods at which nutrients were in excess, declining, and depleted. The effect of nutrient depletion on the phytoplankton community in the absence of enhanced alkalinity was observable in the control treatment. However, it is possible that OAE treatments affected nutrient drawdown during the bloom so that differential nutrient concentrations in the post-bloom phase amplified the emerging differences between the control and OAE treatments."

**Comment 3:** My third reservation is the validity of the three experimental treatments used. Case 1 was a control (no alkalinity change); Case 2 involved an alkalinity increase without equilibration with atmosphere CO2 and so an initial pH > 8.6; and Case 3 involved an alkalinity increase and equilibration with atmospheric CO2, and so an initial pH was closer to the pH of the control. I am concerned about the design of the treatment cases 2 and 3. The goal of this experiment was to assess the effects of alkalinity increase on phytoplankton composition and elemental ratios during photosynthesis. To assess this the initial pH in cases 2 and 3 and the control should have been the same or at least similar. The authors needed to make sure that the effect of pH on phytoplankton composition and elemental uptake ratios in their experiment was minimized. I suspect that the difference in results between Case 1 versus Case 2 or Case 3 may have been because of pH differences rather than the alkalinity differences. In the reported experimental setting, the differences among the three cases might have arisen from pH differences rather than the alkalinity differences or have been affected by their interaction. This is potentially a major problem with the study and interpretation of its results, so the authors should provide appropriate justification for their experimental design.

**Response 3:** Ocean Alkalinity Enhancement entails a variety of changes in the marine carbonate system, even though the increase in alkalinity is its name-giving feature. OAE leads to an increase in pH and decrease in [$CO_2$] with these changes being much more pronounced when not accounting for an immediate invasion of

atmospheric $CO_2$ into the water column (as in the equilibrated treatment). In carbonate chemistry equilibrium, it is impossible to establish the same pH and TA between two treatments when DIC is different between them (Zeebe and Wolf-Gladrow, 2001). As such it is not possible nor the goal of this experiment to manipulate alkalinity independently of pH. We did not report the initial pH of the microcosms (before alkalinity addition) here as it is assumed that all microcosms had the same initial pH as the water was taken from the same location over a ~30-minute time period. This is confirmed by the initial pH of the three control microcosms which varied in pH by 0.01 units. Based on this fact, pH across the three treatments before alkalinity addition was likely the same. The reviewer also states, "In the reported experimental setting, the differences among the three cases might have arisen from pH differences rather than the alkalinity differences or have been affected by their interaction". We agree, it is almost certain that the differences observed in our experiment were due to the divergence in pH and $CO_2$ among microcosms after the addition of alkalinity. Alkalinity itself does not affect biology as it is a chemical concept (Bach et al., 2019a) however the major concern surrounding OAE is the changes in carbonate chemistry associated with the increasing alkalinity (pH, $CO_2$). Assessing the effect of changes in pH and $CO_2$ on a natural phytoplankton community as a result of increased alkalinity is the primary aim of this manuscript. This has been discussed explicitly e.g. in section 4.1.3.

**Comment 4:** My fourth reservation concerns the C:N ratio. The authors measured the C:N ratio throughout the experiment, presumably to enable investigation of the effect of the three treatments on the elemental C:N ratio. If this is the case, only those measurements made under conditions of ample N and P availability are relevant. Once nutrients were depleted, nutrient limitation would more strongly constrain the C:N ratio rather than any change in alkalinity. Although the C:N ratio under the nutrient depleted conditions deviated considerably from the Redfield ratio, its impact is likely to have been minimal as the uptake of C and N by phytoplankton under ample nutrient conditions would have far exceeded those under the depleted conditions.

**Response 4:** We think that it is important to record and show this for the entire experiment and not selectively for only the major bloom phase because plankton communities (and the biogeochemical processes they drive) experience nutrient-depleted conditions for most of the seasonal cycle. Our main argument against the reviewer's reservation is the same as in our reply to comment 2. While C:N stoichiometry may have been affected by nutrient depletion, the temporal differences in nutrient depletion itself was in some cases (i.e. where significant) a treatment effect. Thus, OAE can have influence on C:N indirectly by having first-order effects on nutrient cycling. It is therefore very important to not only consider nutrient-replete stages of a bloom in these types of experiments but also nutrient-deplete stages. To clarify this concern, we will add the following statement to the results where post-bloom C:N data is mentioned: (lines: 401-405) "Differences in the drawdown of inorganic nutrients, particularly $PO_4^3$ and $Si(OH)_4$ (Fig. 4) may have enabled or amplified differences in organic matter stoichiometry, which developed in the post-bloom period. However, it is important to keep in mind that such developments (when significant) were ultimately caused by the treatments, even if they are indirectly induced by direct effects on nutrient drawdown that occurred earlier in the experiment."

**Reviewer #2**

**Comment 5:** What caused the spring bloom in the experiment? In the natural environment the spring bloom is initiated by increasing light availability in a slowly stratifying, nutrient replete water column as day length and mixing length scales change. However, the authors removed these factors from the experiment and placed their microcosms in static irradiance and mixing conditions – what happened in those first few days (1-3 days) in the microcosms? Did all the community acclimate to the new conditions at the same rate? At what point did the composition from flow cytometry analysis between the treatments differ or differentiate? It is a shame that the authors did not assess the compositional changes that occurred at the sampling site over a similar length of time to compare and contrast with their experimental treatments – acknowledging that the sampling site would not have encountered such static conditions.

**Response 5:** We would like to thank the reviewer for highlighting this point. We agree that the spring bloom was most likely caused by the enclosure of nutrient-rich water and exposure to sufficient light to enable a bloom. We acknowledge that enclosure experiments are never capable of representing the full complexity of a real aquatic system, which may compromise the assessment (Carpenter, 1996; see also our response to comment 1 from reviewer 1). From a technical perspective, all microcosms were kept in near-identical conditions, thus the initiation of the spring bloom and the mechanism responsible for this has likely not influenced the outcome of our results and the key message of this manuscript.

Figure 6 of the manuscript illustrates that phytoplankton communities began to diverge approximately two-four days into the experiment. This is with the exception of picoeukaryotes which are known to be significantly affected by $pCO_2$ concentrations, as well as cryptophytes and Synechococcus which are known to be significantly impacted by enclosure (Schulz et al., 2017). We agree that in hindsight sampling of the Derwent Estuary, (location of initial sampling) would have been useful to assess bloom development at the sampling location. Unfortunately, due to the nature of the experiment and timing, there was only one person collecting data each day, thus we were unable to sample from the estuary as well as the microcosms but will undoubtedly consider this in future experiments.

The reviewer also asks the question "Did all the community acclimate to the new conditions at the same rate?". We are not sure how to measure acclimation but based on the flow cytometry data collected we can argue that most if not all assessed groups acclimated at similar times. Figure 6 illustrates an increase in the major phytoplankton groups (picoeukaryotes, nanophytoplankton and microphytoplankton) on day 2. Bacteria and cryptophytes appear to have had a greater starting stock explaining the increasing abundance trend seen for these groups since the initiation of the experiment. Finally, Synechococcus abundance appears to have been declining in response to increasing abundance of other groups, before rapidly declining during the bloom phase. Thus, we believe that the changes observed in the communities at the beginning of the experiment are within normal ranges and follow expected outcomes based on the surplus of nutrients. Furthermore, any differences relative to the natural bloom initiation would not affect our interpretation here as all microcosms were treated in the same manner (apart from the enhancement of alkalinity).

**Comment 6:** Please do not misinterpret my comments, no experiment is without its problems or inherent assumptions and bias. Here the authors need to consider how their treatment of the community may have impacted on the initial dynamics of the organisms present. Rather than discount the observations and insights made, the authors should caveat these in the wider context of the different 'stages' of the experiment and bloom development. Microcosms lasting three weeks are of considerable length, especially considering the 'small' volume involved (starting at 50 L with ~50% removed over time) – do the authors need this entire length of observations to confirm their conclusions about differences in community dynamics through the spring period of replete nutrient drawdown under enhanced light conditions and enhanced alkalinity?

**Response 6:** We would like to first advise the reviewer that although 55 L is indeed a small volume, we removed no more than 15 L from any microcosm over the experimental period for sampling. As we were working with relatively small volumes from the beginning of the experiment, we endeavoured to remove no more than needed each day to minimise potential effects on the phytoplankton community. Bottle effects always affect the outcome in experimental ecological studies, regardless of the size of the container (Carpenter, 1996; Duarte et al., 1997; see also response to comment 1, reviewer 1). Furthermore, to the best of our knowledge, there is no evidence to suggest that a specific duration is an ideal timeframe to minimise bottle effects (Duarte et al., 1997). Three weeks were chosen as we were able to cover a nutrient-induced phytoplankton bloom and the subsequent post-bloom phase with sufficient data points to obtain meaningful insights. We do not deny that bottle effects likely had an impact however bottle effects occur in all micro and mesocosms. To address the reviewers concerns we will add a statement emphasizing this: (also see response to comment 1, reviewer 1)

(Lines 126-132) "Nevertheless, despite some potential advantages, we acknowledge and are fully aware that our microcosm setup cannot reproduce the full physical (or chemical/biological) complexity of nature (Carpenter, 1996). Enclosures of any type will very likely induce so-called bottle effects (Bach and Taucher, 2019), which can alter the observed community succession and therefore affect the transferability of the outcome to natural (non-enclosed) communities (Carpenter, 1996). While this is a general limitation of these kind of experimental studies, we stress that bottle effects would occur in all replicates so that the comparison between control and treatments (as done in our study) is valid."

**Comment 7:** Have the authors considered looking at the particulate C:Si ratios to further elucidate their point about compositional differences in the diatom component of the community as being important between treatments.

**Response 7:** Thank you for highlighting this point. We did look at the stoichiometric ratios between carbon and silica and found no significant differences or discernible patterns between the treatments (see figure below). We thought at the time it would therefore be better to exclude this figure from the article as we already had a large number of figures and did not wish to add any figures depicting insignificant differences. However, we will add

a statement to illustrate that we have considered the ratio of C:Si. (lines: 579-581) "Furthermore, ratios of carbon to silica did not differ between treatments across the experimental period supporting SEM count data (data not shown)".

[Figure]

**Comment 8:** Also, note that Figure A5 is biovolume (um3) rather than biomass and does not make it clear whether there were clear species differences in the communities present – Scanning Electron Microscopy is more than adequate for assessment of compositional differences which would be enlightening in the context of the paper. Maybe some exemplar images could be included to highlight differences or an absence of differences?

**Response 8:** Thank you for highlighting this, we have now changed Figure A5 to Biovolume, not Biomass. We did assess SEM images for compositional differences (peak bloom only) and found little to no differences (see below). At the time of analysis, it was decided that further investigation into the differences was not necessary for this manuscript as our primary goal was to identify the presence or absence of an effect of OAE on a functional type level (not yet in full taxonomic detail, which would have required more resources that were not available to the project). We will clarify this in the discussion: (lines 578-579) "However, there were no clear differences in the composition or biovolume of the diatom community between the control and alkalinity treatments on day 6 (Fig. A5)".

Thank you for the recommendation of including images, we agree that they can be very informative and an appealing way of presenting the information. As we took SEM images to assess biovolume and cell counts/ml many of the images are not (in our opinion) visually appealing to the readers. Furthermore, we have already included five supplementary figures and a supplementary video. As such we decided in this instance not to include any SEM images as we believe they will not add a substantial amount of information to the manuscript. Therefore, we currently have not included any SEM images in the revised manuscript.

[Figure]

**Comment 9:** The manuscript concludes that a deeper assessment of the community and its trophic dynamics is needed to reveal more of the impact and eco-physiological drivers of the responses of the community – this is an important point that should appear clearly in the abstract. Many perturbation experiments simplify their assessment of impact based on generalist bulk perspectives of the community (e.g., chlorophyll, particulate elements) only to conclude that a deeper understanding of the species present is actually needed – it would be beneficial if this was the starting perspective for future studies to ensure that the inner details needed are examined at the right scale.

**Response 9:** We agree and will add the following conclusion to account for this valid concern as the last sentence of the abstract: (lines: 30-31) "We note, however, that more detailed and wide-spread investigations of plankton community responses to OAE are required to confirm or dismiss this first impression."

**Comment 10:** The last line of the abstract is surprising and appears on first reading as a rather controversial conclusion, especially considering the negative impacts on diatom productivity observed and the potential for this to translate into negative impacts for marine ecosystems reliant on their provision of organic matter and essential elements. However, when reading the full manuscript this conclusion was put into much better context – this balanced and fully informed assessment of the statement should appear in the conclusion to ensure that no one reads the abstract (only) and takes home an unbalanced message.

**Response 10:** Thank you for your comment. We agree that this final sentence could be seen as somewhat controversial, especially due to the word "justifiable", which has policy implications. As such we have made subtle changes to the wording of this sentence (line 29) **"Altogether, the inadvertent effects of increased alkalinity on the coastal phytoplankton communities appear to be rather limited relative to the enormous climatic benefit of increasing the inorganic carbon sink of seawater by 21%."** We stand to the point that the climatic benefits of a 21% enlargement of the marine $CO_2$ sink are incredible, and that the effects on the plankton community must therefore be put into perspective.

**References**

Bach, L. T. and Taucher, J.: CO2 effects on diatoms: a synthesis of more than a decade of ocean acidification experiments with natural communities, 15, 1159–1175, https://doi.org/10.5194/os-15-1159-2019, 2019.

Bach, L. T., Gill, S. J., Rickaby, R. E. M., Gore, S., and Renforth, P.: CO2 Removal With Enhanced Weathering and Ocean Alkalinity Enhancement: Potential Risks and Co-benefits for Marine Pelagic Ecosystems, 7, 2019a.

Bach, L. T., Stange, P., Taucher, J., Achterberg, E. P., Algueró-Muñiz, M., Horn, H., Esposito, M., and Riebesell, U.: The Influence of Plankton Community Structure on Sinking Velocity and Remineralization Rate of Marine Aggregates, 33, 971–994, https://doi.org/10.1029/2019GB006256, 2019b.

Carpenter, S. R.: Microcosm experiments have limited relevance for community and ecosystem ecology, Ecology, 77, 677–680, https://doi.org/10.2307/2265490, 1996.

Duarte, C. M., Gasol, J. M., and Vaqué, D.: Role of experimental approaches in marine microbial ecology, 13, 101–111, https://doi.org/10.3354/AME013101, 1997.

Schulz, K. G., Bach, L. T., Bellerby, R. G. J., Bermúdez, R., Büdenbender, J., Boxhammer, T., Czerny, J., Engel, A., Ludwig, A., Meyerhöfer, M., Larsen, A., Paul, A. J., Sswat, M., and Riebesell, U.: Phytoplankton Blooms at Increasing Levels of Atmospheric Carbon Dioxide: Experimental Evidence for Negative Effects on Prymnesiophytes and Positive on Small Picoeukaryotes, Front Mar Sci, 0, 64, https://doi.org/10.3389/FMARS.2017.00064, 2017.

Zeebe, R. E. and Wolf-Gladrow, D.: CO2 in seawater: equilibrium, kinetics, isotopes, Gulf Professional Publishing, 2001.

---

## Author Response (AR2)

Dear referees,

Thank you for your comments on our manuscript. We appreciate the time and effort you have dedicated to providing valuable feedback on our manuscript. Here are our point-by-point responses to your comments.

**Comment 1.** Line 160 – please specify units (PSU)

**Response 1.** In this instance, we measured salinity via the use of electrical conductivity measurements. As such, the salinity value here does not require units as per the international system of units (SI) in oceanography, pages 40 and 44 (UNESCO,1985).

**Comment 2.** Lines 470 and 563 – its suggested that CO2 and H+ are the most "physiologically-relevant carbonate chemistry parameters." I do not necessarily agree with this statement and other parameters are certainly relevant. For example, diatoms use SLC4 bicarbonate transporters (Nakajima et al. 2013). Iron uptake in diatoms can also depend on carbonate (McQuaid et al. 2018).

**Response 2.** We agree that other parameters are indeed relevant. In light of your comment, we will adjust both lines to better reflect this. line 470: "The lower peak chlorophyll $a$ in the equilibrated treatment was unexpected as $CO_2$ and $H^+$, two carbonate chemistry parameters believed to drive phytoplankton growth (Paul and Bach 2020) were relatively similar to the control and within natural ranges (Fig. 4d, e)."
Line 563: "The observation is also remarkable because the differences in BSi occur between the control and both alkalinity treatments even though differences in $CO_2$ and $[H^+]_F$ are much larger between the equilibrated and unequilibrated alkalinity treatments (Figs. 4d, 4e)."

**Comment 3.** Lines 523-525 – How is it known that the largest cells were roughly 50um? From the SEM? Please clarify.

**Response 3.** Yes, this was indeed determined from SEM images. We will adjust these lines to reflect this.
Line 523-525: "The largest diatom cells in our experiment were roughly 50 µm and we did not find any diatoms smaller than 3 µm (determined from SEM). Thus, all diatom cells are most likely found in the nano- and microphytoplankton groups in flow cytometry data."

**Comment 4.** Lines 528-530 – Please clarify what you mean here by "variation in the diatom communities." To me, that means a community shift, but I do not agree that the differences in bSi and Si uptake translate to a shift in the diatom community. I think perhaps some other phrasing might be better here.

**Response 4.** We thank the reviewer for highlighting this. We also agree that the differences observed in our study may not necessarily translate to a shift within the diatom community. Our aim in this sentence was to elucidate that we believe observed differences between treatments (in BSi and Si) indicate some effect and change relating to the diatom community. What this is effect/change is we cannot say with our current data set, in light of your comment we will change this sentence to hopefully better reflect this. Line 529-531: "In addition, significant differences in the build-up of BSi and drawdown of $Si(OH)_4$ between the control and treatments strongly suggests that the alkalinity treatments influenced the diatom communities."

**Comment 5.** Line 571 – The authors suggest that there could be a shift in the diatom community towards non-silicifying species. The only diatom I'm aware of that doesn't use silica is Phaeodactylum tricornutum and it is highly unlikely to have been in these incubations so I don't agree with this conclusion. Or if the authors mean a shift in the phytoplankton community away from diatoms they should clarify that.

**Response 5.** Thank you for highlighting this, we agree that it is highly unlikely that there was a shift in the community towards non-silicifying diatom species. As such we will adjust this sentence to reflect the reviewer's recommendation. Lines 570-571: "For example, there could be a shift in the diatom community towards smaller, less heavily silicified species and/or a higher fraction of non-silicifying phytoplankton."

**Comment 6**. Line 576 and Figure A5 – Rather than stating "data not shown," can the authors please include these data in appendix?

**Response 6.** In consideration of the reviewer's comment, we will include this data in the appendix as follows:

[Figure]

"**Figure A5** Average diatom a) biovolume and b) abundance, during the peak bloom (day 6) within treatments determined via SEM. Data are presented as mean values ± SD."

[Figure]

"**Figure A6** Temporal variation in the molar ratios of TPC to BSi within microcosms. Coloured shading around the respective means represents the standard deviation."

**Comment 7**. Line 695 – A major conclusion is that diatoms were affected due to differences in bSi and Si uptake, but the authors have not considered silicious grazers such as certain Rhizaria. Could those not also influence these results?

**Response 7.** We thank the reviewer for highlighting this very important point. Based on our personal observations and the microscopy conducted during this study we believe that it is highly unlikely that Rhizaria or any other silicifying plankton caused the differences observed in our study. This is because we did not find any silicifiers (except diatoms) in this study, the pilot study we conducted before this study or subsequent sampling from the same location. We do highlight this in line 540 of the text: "Scanning electron microscopy investigations of samples taken before, during, and after the phytoplankton bloom revealed that diatoms were the only silicifiers detected in the plankton community.". We hope that this is sufficient to alleviate the reviewers concerns

**References**

UNESCO (1985) The international system of units (SI) in oceanography, UNESCO Technical Papers No. 45, IAPSO Pub. Sci. No. 32, Paris, France.